# Extracellular matrix stiffness and cell contractility control RNA localization to promote cell migration

Tianhong Wang[1], Susan Hamilla[1,2], Maggie Cam[3], Helim Aranda-Espinoza[2] & Stavroula Mili[1]

Numerous RNAs are enriched within cellular protrusions, but the underlying mechanisms are largely unknown. We had shown that the APC (adenomatous polyposis coli) protein controls localization of some RNAs at protrusions. Here, using protrusion-isolation schemes and RNA-Seq, we find that RNAs localized in protrusions of migrating fibroblasts can be distinguished in two groups, which are differentially enriched in distinct types of protrusions, and are additionally differentially dependent on APC. APC-dependent RNAs become enriched in high-contractility protrusions and, accordingly, their localization is promoted by increasing stiffness of the extracellular matrix. Dissecting the underlying mechanism, we show that actomyosin contractility activates a RhoA-mDia1 signaling pathway that leads to formation of a detyrosinated-microtubule network, which in turn is required for localization of APC-dependent RNAs. Importantly, a competition-based approach to specifically mislocalize APC-dependent RNAs suggests that localization of the APC-dependent RNA subgroup is functionally important for cell migration.

[1] Laboratory of Cellular and Molecular Biology, Center for Cancer Research, National Cancer Institute, NIH, Bethesda, MD 20892, USA. [2] Fischell Department of Bioengineering, University of Maryland, College Park, MD 20742, USA. [3] CCR Collaborative Bioinformatics Resource, National Cancer Institute, NIH, Bethesda, MD 20892, USA. Tianhong Wang and Susan Hamilla contributed equally to this work. Correspondence and requests for materials should be addressed to S.M. (email: voula.mili@nih.gov)

Cell migration is important in a number of physiological processes and disease conditions. During movement, cells asymmetrically extend diverse protrusions towards the front, controlled by the physical properties of the extracellular matrix and the activation status of signaling pathways controlling the actin cytoskeleton. The protrusive front engages in new adhesions with the extracellular matrix, pulling the cell forward[1–3].

It is well appreciated that the formation and maintenance of the polarized state observed in protrusions involves a complex series of interconnected signaling feedbacks[3, 4]. An additional mechanism, however, used in diverse polarized cells, involves asymmetric localization of messenger RNA (mRNA) molecules[5, 6]. Such localized mRNAs contribute to the generation and maintenance of asymmetries mainly through local translation of protein factors[7]. Specific RNAs, RNA-binding proteins, and translation factors are found concentrated at the leading edge and protrusions of migrating cells[8, 9]. Global identification of RNAs from isolated protrusions has revealed a large number of mRNAs that are enriched in protrusions of diverse cell types[10–13]. Significantly, preferentially inhibiting translation at protrusions leads to protrusion destabilization, suggesting that local translation of some of these RNAs is functionally relevant[11]. The exact functional contributions of these localized RNAs, though, are not known.

The mechanisms underlying localization of protrusion-enriched RNAs are poorly understood. There are indications that separate pathways, regulated by distinct RNA-binding proteins, are responsible for targeting RNAs at protrusive regions[14]. The mRNAs encoding β-actin and subunits of the Arp2/3 complex are localized in lamellipodia[8, 15]. Localization and translation of β-actin mRNA is controlled by the ZBP1/IMP1 RNA-binding protein, and interfering with ZBP1 function, or altering its expression levels, affects the distribution of new actin filament nucleation, the directionality of cell migration and the invasiveness of cancer cells[15–17]. An apparently distinct localization pathway relies on the adenomatous polyposis coli (APC) protein. Several RNAs are enriched in protrusions of migrating fibroblasts. These RNAs do not include the β-actin and Arp2/3 subunit mRNAs, and at least some of them require APC for their localization[13]. APC was recently described as a novel RNA-binding protein[18] and associates with protrusion-enriched RNAs in ribonucleoprotein complexes (APC-RNPs)[13]. At the tips of protrusions APC-RNP complexes are anchored at the plus ends of a specific subset of stable microtubules (MTs), which are marked by detyrosination of the alpha-tubulin subunit[13] (termed detyrosinated microtubules or Glu-MTs, because of the penultimate glutamate residue that is exposed upon removal of the C-terminal tyrosine).

Consistent with the local involvement of RNAs in protrusion formation, signaling pathways activated during cell migration control localization of RNAs at protrusions. The Src tyrosine kinase, which is activated upon integrin engagement[19], locally associates with and phosphorylates ZBP1, promoting translation and local production of β-actin[20]. Local activation of the RhoA GTPase, a central regulator of the actin cytoskeleton, is required for localization of β-actin and Arp2/3 subunit RNAs in lamellipodia and for RNA accumulation in protrusions of tumor cells[21–23]. Signaling during cell migration can additionally be mediated by mechanotransduction events, whereby stiffness of the extracellular matrix (ECM) is sensed through mechanosensitive adhesion receptors[24, 25]. Interestingly, applying mechanical tension to cell surface integrin receptors promotes ribosome and mRNA recruitment at the site of stress[26]. However, the exact underlying mechanisms linking mechanical tension to RNA targeting, as well as the identities of the relevant localized RNAs, are not known.

Here we use protrusion-isolation schemes coupled with RNA-Seq to look into the regulation of RNAs enriched in protrusions of fibroblast cells. We show that protrusion-enriched RNAs are not all coordinately regulated, but can be distinguished into two groups based on their dependence on APC. APC-dependent RNAs encode factors involved in processes such as cellular organization and signaling, while APC-independent RNAs encode factors involved in protein synthesis and gene expression and are exemplified by ribosomal protein (r-protein) mRNAs. APC-dependent and APC-independent mRNAs are additionally distinguished by their preferential enrichment in distinct types of protrusions, which appear to exhibit different levels of contractility. Indeed, actomyosin contractility and increased stiffness of the ECM promote localization of APC-dependent RNAs. The underlying mechanism involves activation of a RhoA-mDia1 signaling pathway that promotes formation of a detyrosinated-microtubule network, which in turn is required for localization of APC-dependent RNAs. To address the functional relevance of this mechanism we used a competition-based approach to preferentially mislocalize APC-dependent RNAs. Importantly, this perturbs the efficiency of cell migration on both 2D and 3D substrates. This study describes novel links between mechanosensitive regulation of the microtubule cytoskeleton and RNA localization events important during cell migration.

## Results

**A subset of protrusion-enriched RNAs require APC.** We had previously employed a fractionation scheme that relies on the use of microporous filters to isolate protrusions and cell bodies of fibroblast cells. Using this method, coupled with microarray analysis, we had found that several mRNAs are enriched in protrusive areas. Focusing on a few candidate mRNAs, such as those encoding the Rab13 GTPase or the Pkp4 plakophilin, we had shown that they associate with and depend on the tumor-suppressor protein APC for their localization[13]. Here, to determine whether all RNAs enriched in fibroblast protrusions depend on APC for their localization, we compared control cells to APC knockdown cells, which express ca. 30% of normal APC levels (Supplementary Fig. 1a). Four independent replicates of "protrusion" (Ps) and "cell body" (CB) fractions from each cell type were analyzed by RNA-Seq. Principal component analysis showed that replicates cluster together, indicating good reproducibility (Supplementary Fig. 1b). Scatter plots of average expression values showed that most RNAs were equally present in Ps and CB fractions (Supplementary Fig. 1c). To identify RNAs localized at protrusions, we calculated a Ps/CB enrichment ratio. RNAs that were significantly enriched in protrusions of control cells (fold change (FC) > 2, $p$-value < 0.05, Methods section) were considered Ps-enriched and they corresponded to ca. 7% of detected RNAs (914 of 13015 RNAs) (Fig. 1a and Supplementary Data 1). To determine how enrichment of these RNAs was affected by APC knockdown, we assessed the $\log_2$ FC difference between control and APC knockdown cells, $\log_2(\text{Diff FC}) = \log_2(\text{FC}_{\text{APCKD}}) - \log_2(\text{FC}_{\text{control}})$. Positive values indicate increased enrichment, while negative values indicate decreased enrichment upon APC knockdown. As seen in Fig. 1b, a subset of Ps-localized RNAs became less enriched upon APC knockdown shifting the cumulative fraction plot to negative values (red line; "Ps-enriched in control"). Interestingly, the remaining Ps-localized RNAs were not affected, with $\log_2$Diff FC values centering around zero. These results indicate the existence of distinct RNA subsets that differentially depend on APC for their localization at protrusions. By contrast, $\log_2$Diff FC values of non-localized RNAs predominantly centered around zero

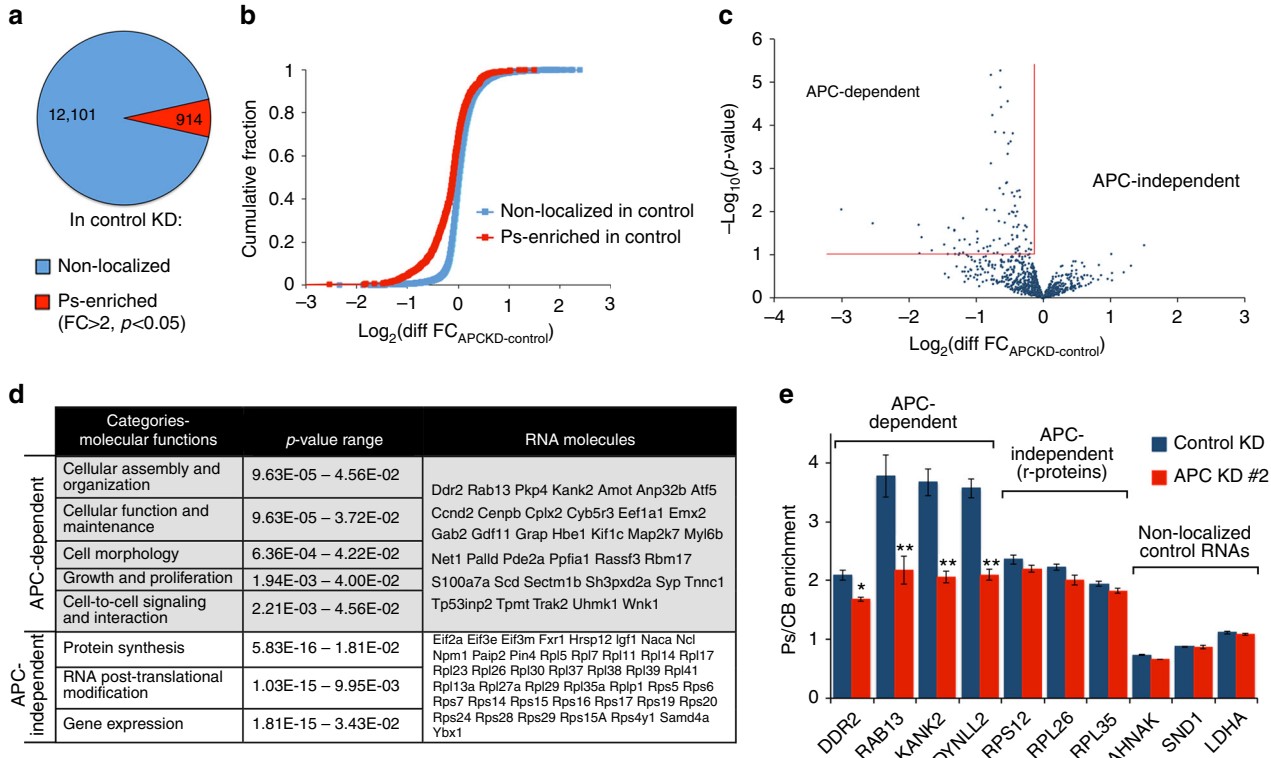

**Fig. 1** Distinct RNA groups are targeted at protrusions through APC-dependent and APC-independent pathways. **a** Pie chart of non-localized or protrusion (Ps)-enriched RNAs in control knockdown (KD) cells from 4 replicate experiments. Cutoffs for Ps-enrichment were set at FC > 2 and p-value < 0.05. **b** Cumulative fraction plot of log2 fold-change differences between control and APC knockdown cells, for RNAs non-localized or localized in protrusions of control cells. **c** For all Ps-localized RNAs of control cells, the $\log_2$ differences in FC values between control and APC knockdown cells were plotted against the corresponding p-values. Applying the cutoffs marked by the red lines (see Supplementary Data 1 for details), Ps-localized RNAs were distinguished into APC-dependent and APC-independent groups. **d** Categories of molecular functions, derived through IPA analysis, significantly represented in APC-dependent and APC-independent RNA groups. The identities of the RNAs included in the groups are listed. **e** Control or APC knockdown cells were fractionated into Ps and cell body (CB) fractions. The indicated RNAs were detected through nanoString analysis to calculate Ps/CB enrichment ratios (n = 3; error bars: standard error). p-value: **< 0.0001, *< 0.04 by analysis of variance with Bonferroni's multiple comparisons test against the corresponding control

(blue line; "Non-localized in control"), indicating that APC knockdown does not result in considerable overall changes in RNA distributions, and underscoring the specificity of the observed effects (Fig. 1b).

We applied stringency cutoffs based on $\log_2$(Diff FC) and p-value (Fig. 1c, red lines and detailed in Supplementary Data 1) in order to distinguish Ps-localized RNAs into APC-dependent and APC-independent groups. These groups contain 91 and 821 RNAs respectively (Supplementary Data 1), and we examined the functional categories associated with each group. Ingenuity pathway (IPA) analysis revealed that APC-dependent and APC-independent RNAs are associated with quite distinct functional classes (Fig. 1d and Supplementary Data 2). APC-dependent RNAs are enriched for factors involved in molecular processes such as cellular assembly, organization, morphology, growth, and cell-to-cell signaling. RNAs represented in these categories include *Pkp4*, *Rab13*, *Ddr2*, and *Kank2*, consistent with our prior studies identifying them as being dependent on APC[13]. On the other hand, APC-independent RNAs are enriched for factors involved in protein synthesis and gene expression (Fig. 1d and Supplementary Data 2). Among APC-independent RNAs, a prominent group corresponds to r-protein mRNAs, which encode the ca. 80 protein constituents of the 40S and 60S ribosomal subunits. Indeed, virtually all r-protein mRNAs were enriched > 1.5 fold in fibroblast protrusions (Supplementary Data 1). We have been focusing on r-protein mRNAs as representatives of the APC-independent localized group.

To validate the existence of APC-dependent and APC-independent RNA groups and to rule out any contribution of bias introduced by complementary DNA (cDNA) generation and amplification during RNA-Seq library preparation, we additionally analyzed Ps and CB RNAs using the nCounter nanoString technology that allows direct RNA detection and counting without amplification. We included probes to detect all r-protein mRNAs (79 of them were reliably detected), 10 selected APC-dependent RNAs as well as a set of non-localized control RNAs (Fig. 1e shows representative mRNAs of each group and Supplementary Fig. 2 the complete data set). The control RNAs were selected because of their behavior as diffusely distributed RNAs, at least in NIH/3T3 fibroblasts, based on our prior RNA-Seq, microarray and in situ hybridization analyses. Consistent with the results described above, both the r-protein and the candidate APC-dependent RNAs exhibited a higher degree of enrichment in protrusions compared to the non-localized controls. Furthermore, APC knockdown preferentially reduced the enrichment of all APC-dependent RNAs tested, while minimally affecting the enrichment of r-protein mRNAs in protrusions (Fig. 1e and Supplementary Fig. 2). Similar results were obtained when knocking down APC using small interfering RNAs (siRNAs) targeting a different region (Supplementary Fig. 3a, b), indicating that the observed effects are indeed due to loss of APC expression. These results, thus, validate the existence of two distinct groups of Ps-localized RNAs, which are distinguished by their dependence on APC.

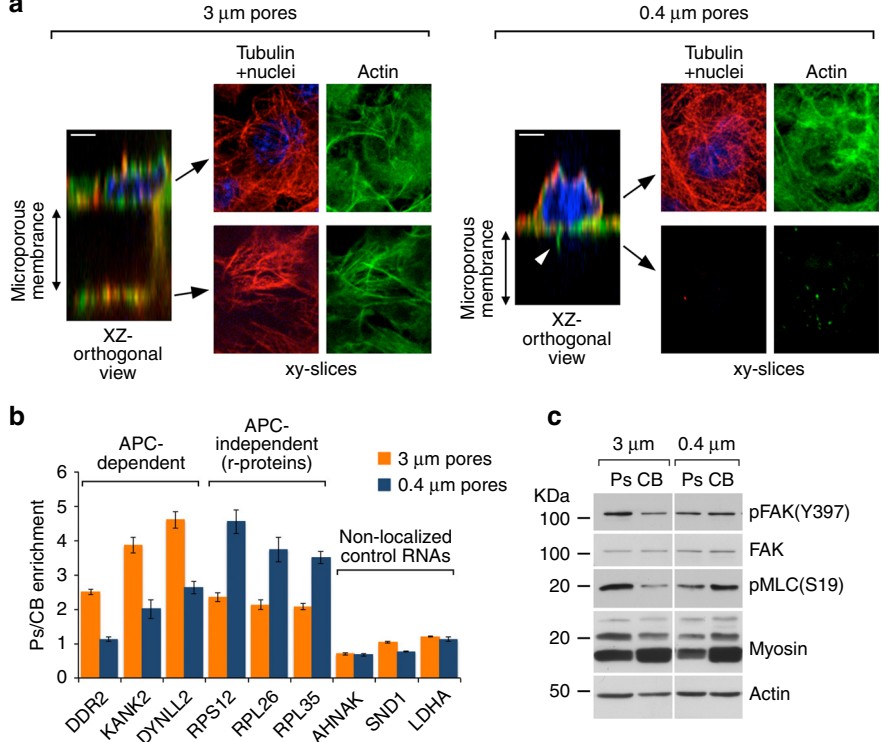

**Fig. 2** APC-dependent and APC-independent RNAs are enriched in different types of protrusions, associated with high or low levels of contractility. **a** NIH/3T3 fibroblast cells were plated on microporous filters containing pores of 3 or 0.4 µm in diameter. The cells were induced to extend protrusions by addition of LPA in the bottom chamber and were subsequently fixed and stained to detect actin (green), tubulin (red), and nuclei (DAPI, blue). Confocal xy-slices through the cell body or protrusions, or xz-orthogonal views are shown. Scale bars: 4 µm. **b** Ps and CB fractions were isolated from cells extending protrusions through 3 or 0.4 µm pores. The indicated RNAs were detected through nanoString analysis to calculate Ps/CB enrichment ratios ($n = 3$; error bars: standard error). **c** Ps and CB fractions, of cells extending protrusions through 3 or 0.4 µm pores, were analyzed by western blot to detect the indicated proteins. Results are representative of two independent repeats

**Distinct RNA enrichment patterns in different protrusions**.
Examination of the above data sets revealed another distinction between the two localized RNA groups. APC-dependent RNAs were consistently enriched at protrusions to a higher degree than the APC-independent r-protein mRNAs. This difference is observed both for the select RNAs detected in the nanoString analysis (Fig. 1e and Supplementary Figs. 2 and 3a) as well as for the complete groups derived from RNA-Seq (Supplementary Fig. 3c). This pattern of RNA enrichments was observed in protrusions extended through pores of 3 µm in diameter. Under these conditions, the cells extend long protrusions through the microporous filter that reach and spread on its undersurface. These protrusions contain well-developed cytoskeletal structures, including microtubules and actin cables (Fig. 2a, left panels).

To test whether the type of protrusion would have an impact on the pattern of enriched RNAs, we isolated protrusions from cells plated on filters containing pores of 0.4 µm in diameter. In this case, the cells extend smaller protrusions that do not reach the undersurface. These protrusions contain actin but relatively few microtubules and are reminiscent of filopodia (Fig. 2a, right panels). Analysis of RNA enrichment in these filopodia-like protrusions strikingly revealed a reversed RNA localization pattern. Specifically, r-protein mRNAs were now enriched to a higher degree while enrichment of APC-dependent RNAs was reduced, in contrast to the distribution exhibited in the larger protrusions (Fig. 2b shows representative RNAs and Supplementary Fig. 4 the complete data set). These data show that APC-dependent and APC-independent RNAs differ not only in terms

of their dependence on APC but are also preferentially enriched in distinct types of protrusions.

**Myosin levels correlate with APC-dependent RNA localization**.
We further sought to understand what feature might underlie the difference in enrichment of APC-dependent RNAs. As previously observed, the larger protrusions extending through the 3 µm pores are enriched in activated FAK, phosphorylated at Y397 (pFAK(Y397)) (Fig. 2c)[13, 27]. This finding is consistent with the formation of new focal adhesions, which can in turn anchor contractile actomyosin stress fibers. Indeed, abundant actin bundles are observed in these protrusions (Fig. 2a) and western blot (WB) analysis shows an enrichment of activated, phosphorylated myosin light chain (pMLC) associated with higher contractile activity (Fig. 2c)[28]. By contrast, the protrusions extending through the 0.4 µm pores do not show an enrichment of pFAK(Y397) and are rather depleted both for myosin as well as for pMLC (Fig. 2c). These results, thus, indicate that the large and small protrusions might exhibit different levels of contractility. APC-dependent RNAs are enriched in protrusions that apparently exhibit increased contractility, while their enrichment is reduced in the likely less contractile, filopodia-like protrusions.

**Substrate stiffness promotes APC-dependent RNA localization**.
To test whether, indeed, actomyosin contractility is responsible for the increased accumulation of APC-dependent RNAs at protrusions, we used fibronectin-coated polyacrylamide gels of different stiffness. On softer substrates, fibroblasts exert less

traction force and are less contractile than on stiffer substrates[29], [30]. Indeed, NIH/3T3 cells plated on soft, fibronectin-coated 1 kPa gels, are mostly round with a small spreading area (Fig. 3b and Supplementary Fig. 5a)[31]. However, when they are plated on gels of intermediate (5 kPa) or high (280 kPa) stiffness, the spreading area increases concomitant with the formation of longer protrusions (Fig. 3b and Supplementary Fig. 5a). Additionally, measurement of traction forces confirms that on substrates of increased stiffness the cells exert higher traction forces reflecting higher contractility (Fig. 3c).

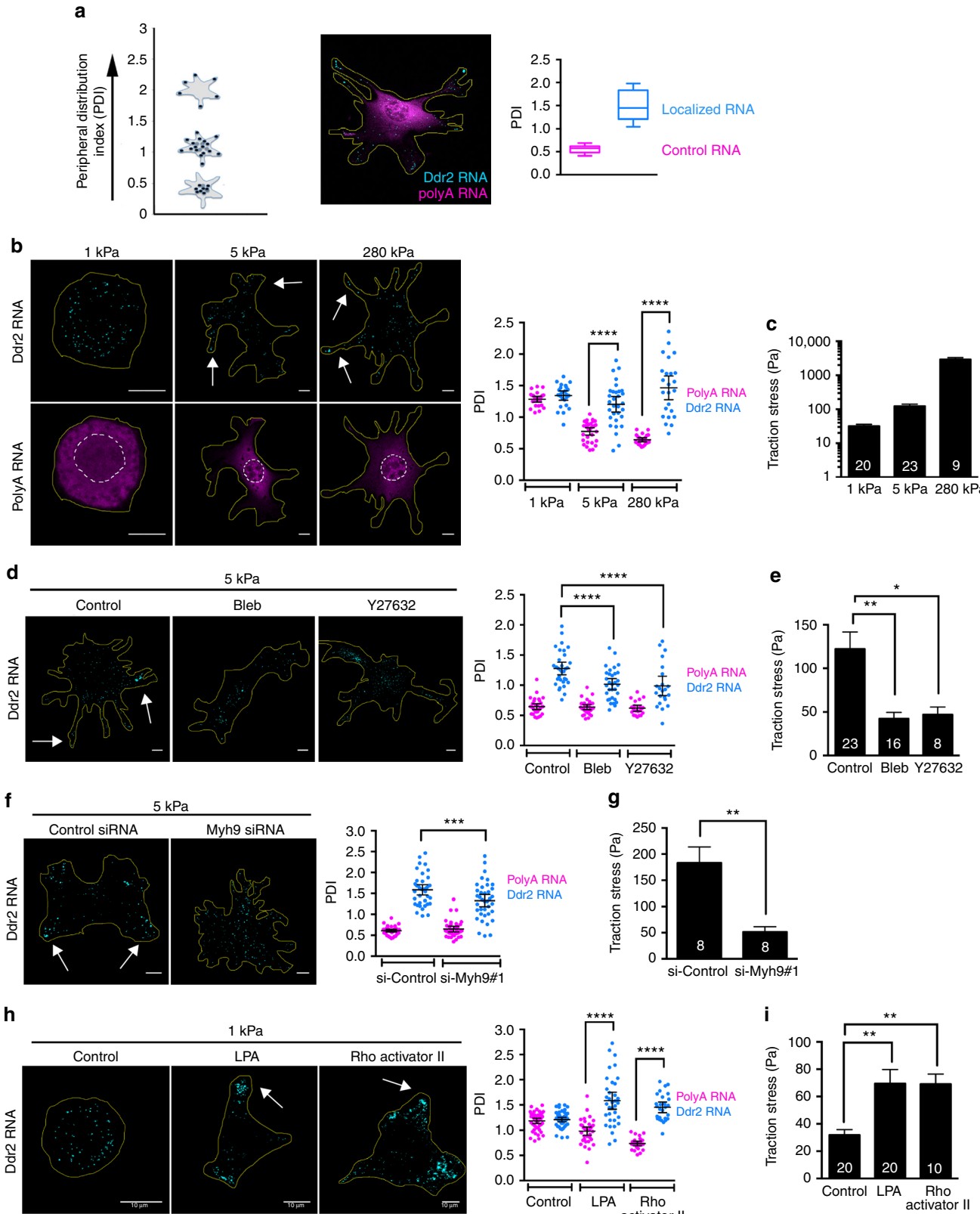

To visualize RNA distributions, we used high-resolution fluorescence in situ hybridization. As we had previously reported, APC-dependent RNAs, such as *Ddr2*, *Pkp4*, and *Rab13* are observed accumulating at the cell periphery or at the tips of protrusions in cells grown on two-dimensional glass surfaces[13, 32]. We focused on the *Ddr2* RNA, encoding the Discoidin domain collagen receptor 2, as a representative APC-dependent RNA. As internal controls, we detected either specific, diffusely distributed RNAs (*Arpc3*, *RhoA*), or the overall distribution of polyadenylated RNAs (polyA RNA), visualized through oligo(dT) hybridization. Furthermore, to quantitatively assess RNA distributions in populations of cells, we developed a method, adapted from Park et al.[33], to derive a metric termed the peripheral distribution index (PDI) (Fig. 3a). This index is derived by calculating the second moment of RNA pixel intensity positions relative to the centroid of the nucleus. To account for differences in cell shape, the second moment of RNA is normalized to the second moment of a hypothetical uniform distribution, derived from a binary mask image of each cell. Therefore, PDI = 1 indicates a uniform, diffuse signal, PDI < 1 a perinuclear, central signal, while PDI > 1 indicates a peripheral signal (Fig. 3a). We note that in practice, when detecting polyA RNA, the signal within the nuclear area is subtracted prior to calculation of PDI values. This creates an apparent increase in PDI, which is more pronounced in cells with small spreading area (see Methods section for details and additional considerations for PDI analysis). Overall, PDI assessment can distinguish diffusely distributed from peripherally localized RNAs within a population of cells (Fig. 3a).

Assessment of *Ddr2* RNA distribution on substrates of varying stiffness revealed that, on soft (1kPa) substrates, the *Ddr2* RNA was not localized, but was distributed similar to the overall polyA RNA population, as evidenced by the comparable PDI indices (Fig. 3b). On intermediate (5 kPa) or stiff (280 kPa) substrates, polyA RNA distribution became more perinuclear (reflecting the larger cytoplasmic volume around the nucleus) with only a small proportion reaching into the protrusive regions (which contain a thinner layer of cytoplasm). This change in polyA RNA distribution is reflected in the lower PDI values (Fig. 3b). By contrast, the *Ddr2* RNA was preferentially localized towards the peripheral protrusions or accumulated at their tips (Fig. 3b), reflected in the higher PDI values compared to the PDI values of polyA RNAs in the same cells. We have observed similar results for the APC-dependent RNAs *Pkp4* and *Cyb5r3*, but not for the non-localized *RhoA* RNA (Supplementary Fig. 5b, c). Furthermore, plating on different substrates does not appreciably change the expression levels of APC-dependent RNAs (Supplementary Fig. 5d). Therefore, increased substrate stiffness enhances the peripheral localization of APC-dependent RNAs.

To directly test the role of actomyosin contractility on APC-dependent RNA localization, we used pharmacological and RNAi approaches to manipulate either myosin II activity or the RhoA-ROCK pathway, which promotes cellular contractility through myosin light chain phosphorylation[34]. Inhibition of myosin II, upon blebbistatin treatment, or inhibition of ROCK, using the specific inhibitor Y27632, resulted in reduced cellular contractility, on substrates of intermediate stiffness, as assessed through traction force microscopy (Fig. 3e). Importantly, this effect was accompanied by a significant reduction in the peripheral localization of *Ddr2* RNA (Fig. 3d). Similar reduction in traction stress and *Ddr2* RNA localization was observed upon knockdown of the myosin heavy chain IIA, Myh9, using different siRNAs (Fig. 3f, g and Supplementary Fig 6a).

We further tested whether, conversely, inducing contractility of cells on soft 1 kPa substrates would be sufficient to promote RNA localization at protrusions. For this, we used two different ways of activating RhoA to stimulate actomyosin contraction. Treatment with lysophosphatidic acid (LPA) or treatment with the Rho Activator II, which is based on a bacterial toxin and induces RhoA activation by blocking its intrinsic and GAP-stimulated GTPase activity. Indeed, in both cases, treated cells became more elongated, increased their spreading area and exerted more traction stress compared to control cells (Fig. 3i and Supplementary Fig. 5e). Significantly, LPA treatment as well as treatment with RhoA Activator II preferentially increased the peripheral accumulation of *Ddr2* RNA compared to polyA RNA (Fig. 3h). Taken together, these data indicate that RhoA activation and actomyosin contractility are both necessary and sufficient to promote APC-dependent RNA localization at cell protrusions.

**ECM stiffness and actomyosin contractility regulate Glu-MTs.** While the above data reveal a requirement for actomyosin contractility, our previous studies indicated that APC-dependent RNAs associate with the microtubule cytoskeleton and specifically with stable Glu-MTs. We thus investigated whether changes in contractility affect Glu-MTs. Less contractile cells, on soft 1 kPa substrates, had barely any visible Glu-MTs, despite the presence of an obvious microtubule network (Fig. 4a). By contrast, a large proportion of the more contractile cells, on 5 or 280 kPa, exhibited well-defined Glu-MTs (Fig. 4a). A similar increase in Glu-tubulin levels as substrate stiffness increases was also detected by WB (Supplementary Fig. 6b). Furthermore, increasing or decreasing contractility led to corresponding changes of the Glu-MT network. Specifically, on substrates of intermediate or high stiffness, treatment with blebbistatin or Y27632, decreased or abolished Glu-MTs in the cell body and overall reduced the number of cells with developed Glu-MT network (Fig. 4b). On the other hand, formation of Glu-MTs was induced on soft substrates upon treatment with LPA or Rho Activator II (Fig. 4c). We thus conclude that actomyosin contractility is required for

**Fig. 3** Increased substrate stiffness and actomyosin contractility enhance APC-dependent RNA localization. **a** Left panel: schematic depicting changes in PDI index values for the indicated hypothetical RNA distributions. Higher PDI values reflect more peripheral RNA distributions. Middle panel: in situ hybridization of Ddr2 RNA (localized, cyan) and polyA RNA (control, magenta) in a representative cell on glass. Right panel: corresponding PDI values from a population of cells. Note that, in order to compare cytoplasmic distributions, signals within the nuclear area are subtracted prior to calculation of PDI values. This creates an apparent increase in PDI, which is disproportionately observed on 1kPa substrates (e.g., in panels **b**, **h**), because of the larger relative area covered by the nucleus. **b** Representative Ddr2 and polyA RNA distributions on substrates of varying stiffness, and corresponding PDI values. Arrows point to peripheral Ddr2 RNAs. **c** Traction force microscopy analysis on substrates of the indicated stiffness. **d** Representative Ddr2 RNA distributions and PDI values of cells treated with blebbistatin (Bleb) or Y27632, plated on 5 kPa substrates. **e** Traction force microscopy analysis of cells treated as in **d**. **f** Representative Ddr2 RNA distributions and PDI values of control cells or cells knocked-down for Myosin heavy chain IIA, Myh9, plated on 5kPa substrates. **g** Traction force microscopy analysis of cells treated as in **f**. **h** Representative Ddr2 RNA distributions and PDI values of control cells or cells treated with LPA or Rho activator II, plated on 1 kPa substrates. **i** Traction force microscopy analysis of cells treated as in **h**. For all PDI quantifications, bars represent the mean with 95% confidence interval. Points indicate individual cells analyzed in at least two or more independent experiments. For all traction force analyses, error bars represent standard error and *n*-values are indicated within each bar. *p*-values: ****<0.0001, ***<0.001, **<0.01, *<0.05 by analysis of variance (**b–f**, **h**, **i**) or Mann–Whitney test (**g**). Scale bars: 10 μm. Yellow lines: cell outline; white dashed line: nuclear outline

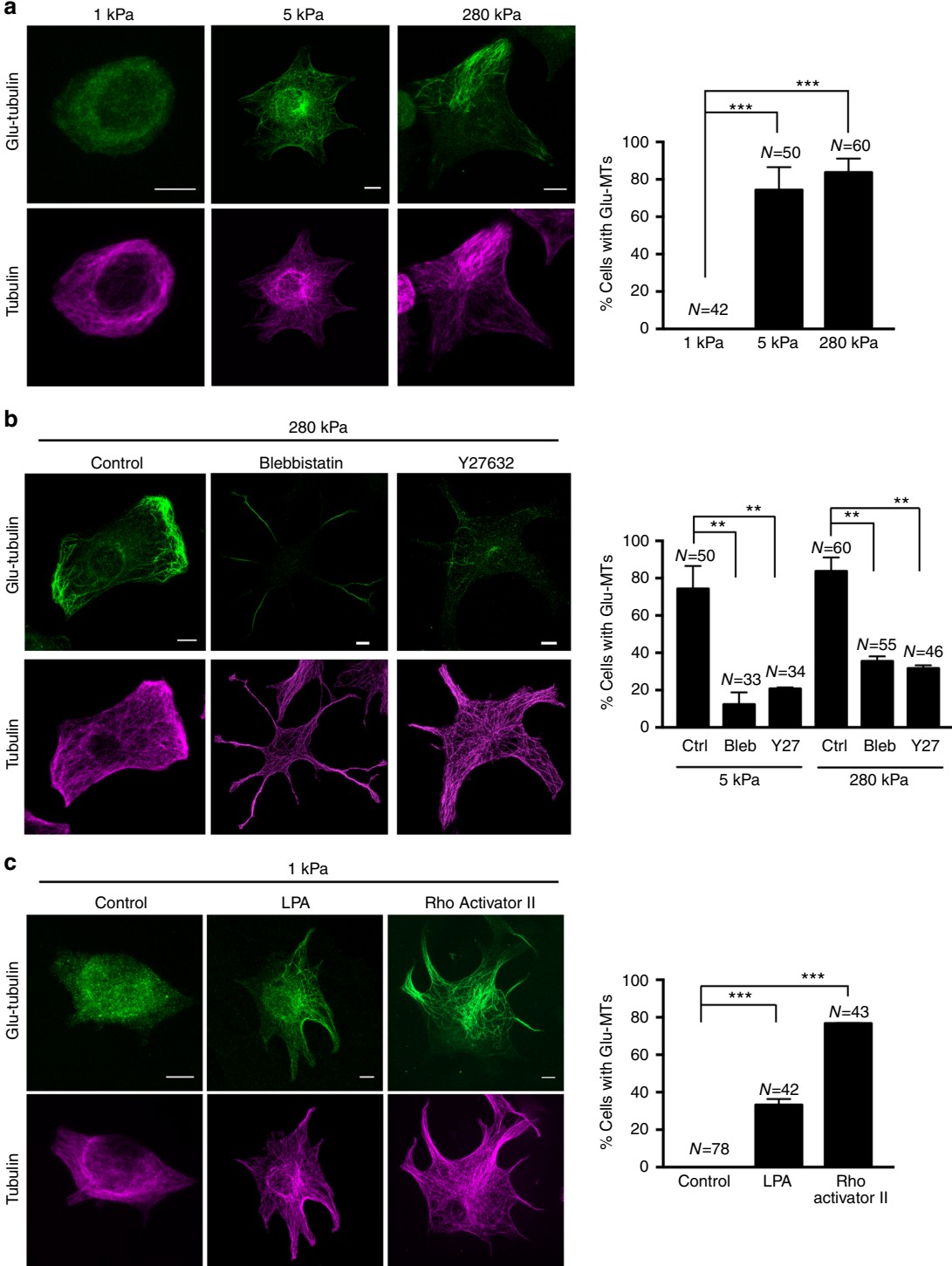

**Fig. 4** Actomyosin contractility is required for formation of the detyrosinated-microtubule network. Representative immunofluorescence staining images of Glu-tubulin or total alpha-tubulin. Graphs show corresponding percentages of cells exhibiting a Glu-MT network. **a** Cells were plated on substrates of the indicated stiffness. **b** Cells on 5 or 280 kPa substrates were treated with Blebbistatin or Y27632. We note that cells treated with Blebbistatin mostly retain Glu-tubulin staining within their long, thin protrusions. Cells were scored for the presence of Glu-MTs within the main cell body. **c** Cells on 1 kPa substrates were treated with LPA or Rho activator II. Error bars: standard error. N: total number of cells observed in two or more biological replicates. p-values: ***< 0.001, **< 0.01 by analysis of variance with Bonferroni's multiple comparisons test. Scale bars: 10 μm

formation of the detyrosinated-MT network and concomitantly affects the localization of APC-dependent RNAs.

**Contractility regulates RNA localization through Glu-MTs.** To understand whether the effect on Glu-MTs might be causatively

connected to RNA localization, we modulated Glu-MT levels by manipulating the enzymes involved in their formation. Glu-MTs are formed upon removal of the C-terminal tyrosine of α-tubulin by tubulin carboxypeptidase (TCP) (Fig. 5a). Even though the gene encoding TCP is unknown, the chemical compound

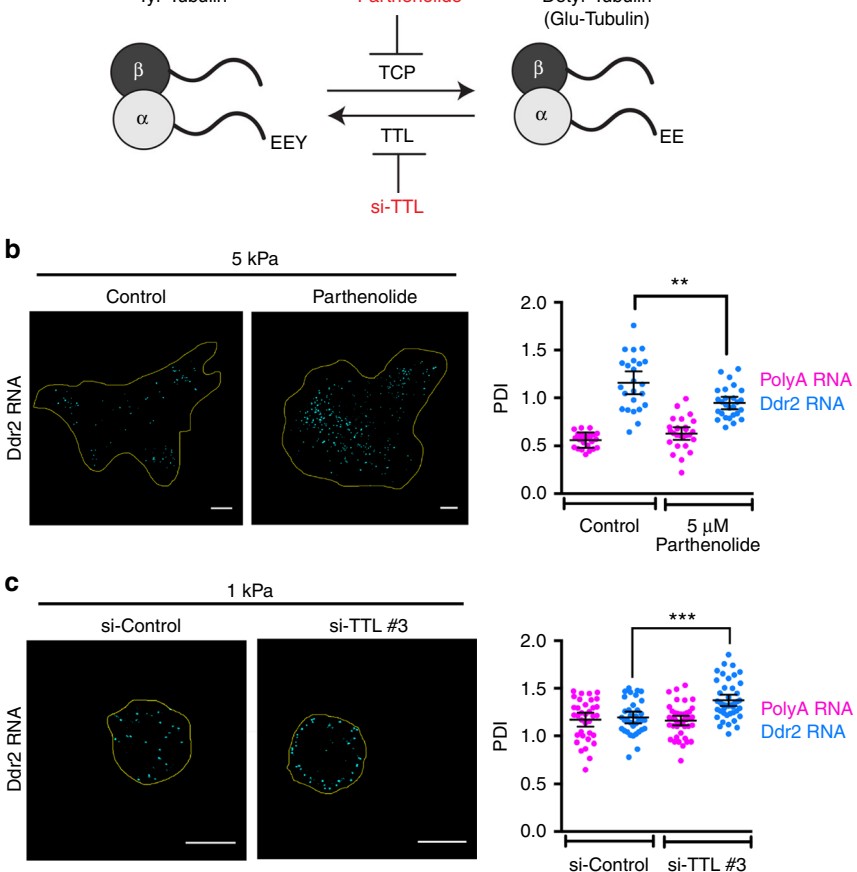

**Fig. 5** Detyrosinated microtubules are necessary and sufficient for the peripheral localization of Ddr2 RNA. **a** Schematic depicting the microtubule detyrosination cycle. Indicated is the C-terminal tyrosine residue removed by an unknown carboxypeptidase. The effects of parthenolide treatment or knockdown of tubulin tyrosine ligase (TTL) are also shown. **b** FISH analysis of control or parthenolide treated cells plated on 5 kPa substrates. Representative Ddr2 RNA distributions and PDI values of Ddr2 and polyA RNA are shown. **c** FISH analysis of control or TTL knockdown cells plated on 1 kPa substrates. Representative Ddr2 RNA distributions and PDI values of Ddr2 and polyA RNA are shown. For PDI quantifications, bars represent the mean with 95% confidence interval. Points indicate individual cells analyzed in at least two or more independent experiments. *p*-values: ***< 0.001 by analysis of variance with Bonferroni's multiple comparisons test. Scale bars: 10 μm

Parthenolide inhibits TCP activity and reduces formation of Glu-MTs[35] (Supplementary Fig. 6c). The action of TCP can be reversed by tubulin tyrosine ligase (TTL), which adds back the C-terminal Tyrosine. Knockdown of TTL, thus, increases the amount of Glu-MTs (Supplementary Fig. 6c). Contractile cells plated on 5 or 280 kPa substrates, briefly treated with Parthenolide showed a decrease in peripheral *Ddr2* RNA localization (Fig. 5b), indicating that Glu-MTs are necessary for localization of the *Ddr2* RNA. On the other hand, TTL knockdown cells plated on soft 1 kPa substrates exhibited a significant increase in peripheral *Ddr2* accumulation (Fig. 5c and Supplementary Fig. 6d). While it is likely that additional factors are required for APC-dependent RNA localization, these results suggest that Glu-MTs are necessary and, in certain cases, such as under low contractility conditions, are limiting for peripheral RNA localization.

Importantly, blocking microtubule detyrosination with Parthenolide suppressed the ability of LPA to enhance RNA localization on soft substrates (Fig. 6a). Furthermore, knocking down TTL, rescued detyrosinated microtubules in the presence of blebbistatin (Fig. 6c) and importantly, also rescued peripheral RNA localization in blebbistatin-treated cells (Fig. 6b). These data strongly suggest that actomyosin contractility affects APC-dependent RNA localization through impacting on the formation of the

Glu-MT network, which in turn is necessary for the localization and/or anchoring of these RNAs at protrusions.

A factor implicated in Glu-MT regulation downstream of RhoA is the formin mDia[36, 37]. To assess the role of mDia1 in the mechanism connecting actomyosin contractility to APC-dependent RNA localization, we expressed a truncated mutant of mDia, GFP-mDia-(ΔN3), which lacks the N-terminal autoinhibitory domain and thus is constitutively active[38]. Expression of GFP-mDia-(ΔN3) in already contractile cells (5 kPa) resulted in a tendency to form parallel Glu-MT bundles, and only slightly increased the percentage of cells exhibiting a Glu-MT network without changing the peripheral distribution of the *Ddr2* RNA as assessed by its PDI index (Fig. 6d, e). However, in cells treated with blebbistatin to reduce contractility, GFP-mDia-(ΔN3) expression was sufficient to rescue both the formation of Glu-MTs as well as *Ddr2* RNA localization compared to cells treated with blebbistatin alone (Fig. 6d, e). These results indicate that actomyosin contractility regulates Glu-MTs and APC-dependent RNA localization likely through affecting the activity of mDia1.

**A competition scheme to mislocalize RNAs from protrusions.** To address the functional role of APC-dependent RNA localization, we used a competition-based approach. We had

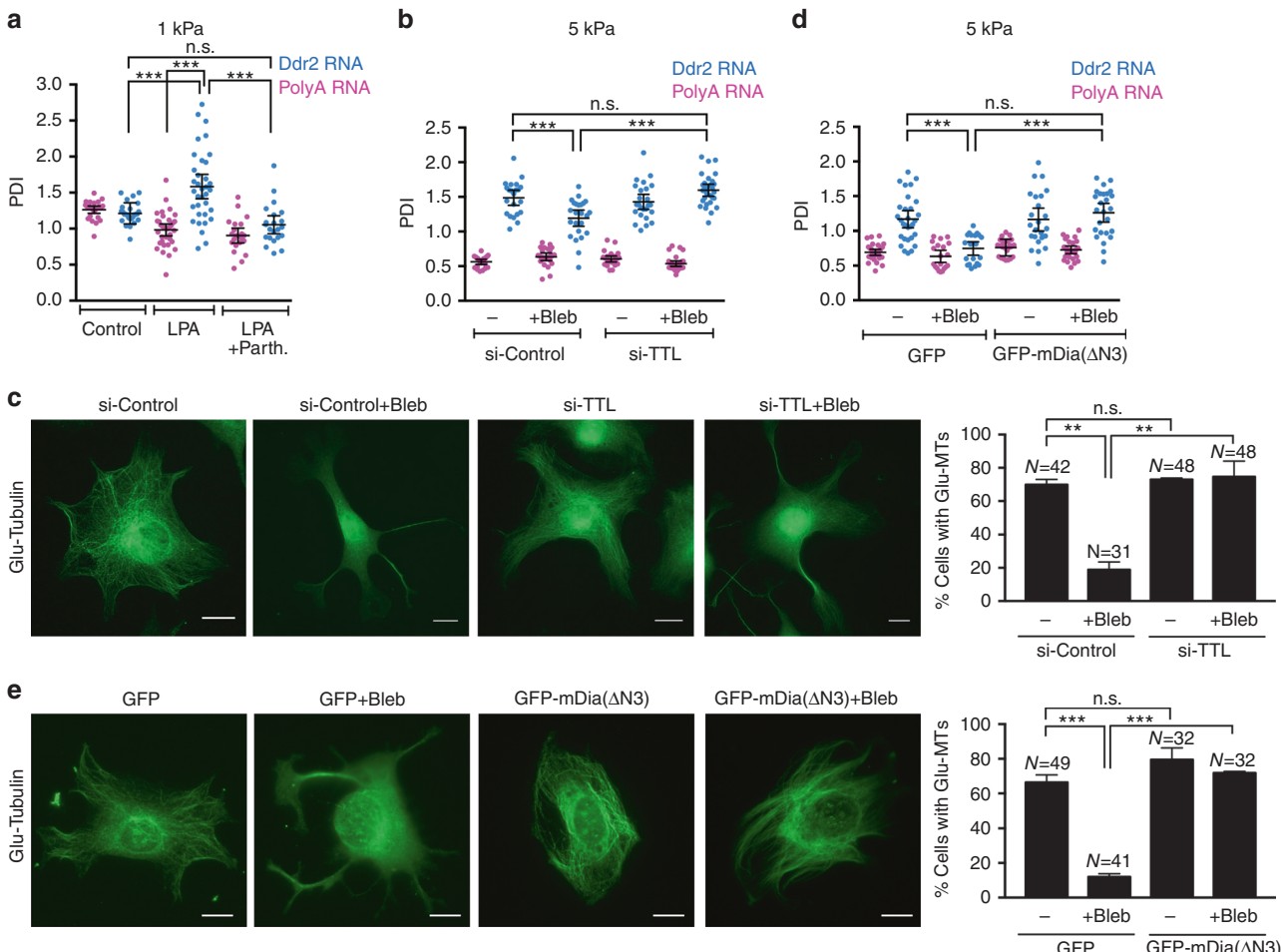

**Fig. 6** Detyrosinated microtubules mediate the effect of contractility on APC-dependent RNA localization at protrusions. **a** FISH analysis of cells on 1 kPa substrates treated with LPA alone or LPA together with parthenolide. PDI values of Ddr2 and polyA RNA are shown. **b** FISH analysis of control or TTL knockdown cells, treated or not with blebbistatin as indicated. Cells were plated on 5 kPa substrates. PDI values of Ddr2 and polyA RNA are shown. **c** Representative Glu-tubulin immunofluorescence staining of cells treated as in **b**. Graph shows corresponding percentages of cells exhibiting a Glu-MT network. **d** FISH analysis of cells transfected with GFP or GFP-mDia(ΔN3), treated or not with blebbistatin as indicated. Cells were plated on 5 kPa substrates. PDI values of Ddr2 and polyA RNA are shown. **e** Representative Glu-tubulin immunofluorescence staining of cells treated as in **d**. Graph shows corresponding percentages of cells exhibiting a Glu-MT network. For all PDI quantifications, bars represent the mean with 95% confidence interval. Points indicate individual cells analyzed in at least two or more independent experiments. For cell percentages, error bars show standard error and N is the total number of cells observed in two or more biological replicates. p-values: ***< 0.001, **< 0.01 by analysis of variance with Bonferroni's multiple comparisons test. Scale bars: 10 μm

previously shown that the 3′ untranslated regions (UTRs) of the *Pkp4* or *Rab13* transcripts were sufficient to target the non-localized *β-globin* RNA at protrusions, indicating that they contain a "protrusion-targeting" element[13]. Through testing deletion fragments, we now find that a 5′ region of the *Pkp4* 3′ UTR (fragments PkpA12 and PkpA12.1), retain protrusion-localization activity (albeit not to the full extent), while a fragment encompassing the 3′ region (PkpB) was not able to direct *β-globin* RNA localization at protrusions (Fig. 7a and Supplementary Fig. 7a, b). These constructs were placed under a doxycycline-inducible promoter, and stably transduced fibroblast cell lines were derived. Expression was induced to > 60 fold over the corresponding endogenous *Rab13* or *Pkp4* RNAs (Supplementary Fig. 7c), with the expectation that high expression of localization-competent mRNAs would compete with and mislocalize the endogenous APC-dependent RNAs. Such competing UTR (cUTR) lines (expressing the β-globin coding sequence followed by the Rab13, Pkp4, PkpA12, or PkpA12.1 UTRs) were compared to the control lines (expressing the β-globin coding sequence followed by the β-globin UTR (HBB) or

the non-localized PkpB UTR fragment). These non-localized constructs serve to control for non-specific effects that might be mediated through sequestration of RNA-binding factors unrelated to localization. The rationale being that effects mediated through titration of unrelated factors would not be expected to correlate with the presence of a protrusion-targeting signal within the overexpressed construct.

To determine whether cUTR expression affects APC-dependent RNA distribution, we used these lines to first assess the distributions of the endogenous *Ddr2* and *Kank2* RNAs, after in situ hybridization. Indeed, cell lines expressing either the full length Rab13 or Pkp4 UTRs, or the localized Pkp4-fragments, exhibited a significant reduction in the peripheral localization of both *Ddr2* and *Kank2* RNAs (Fig. 7b). Importantly, the non-localized control UTRs did not have that effect. Thus, cUTRs can successfully mislocalize endogenous APC-dependent RNAs.

To further determine whether cUTR expression specifically and more generally mislocalized APC-dependent RNAs, we isolated four independent replicate samples of protrusions

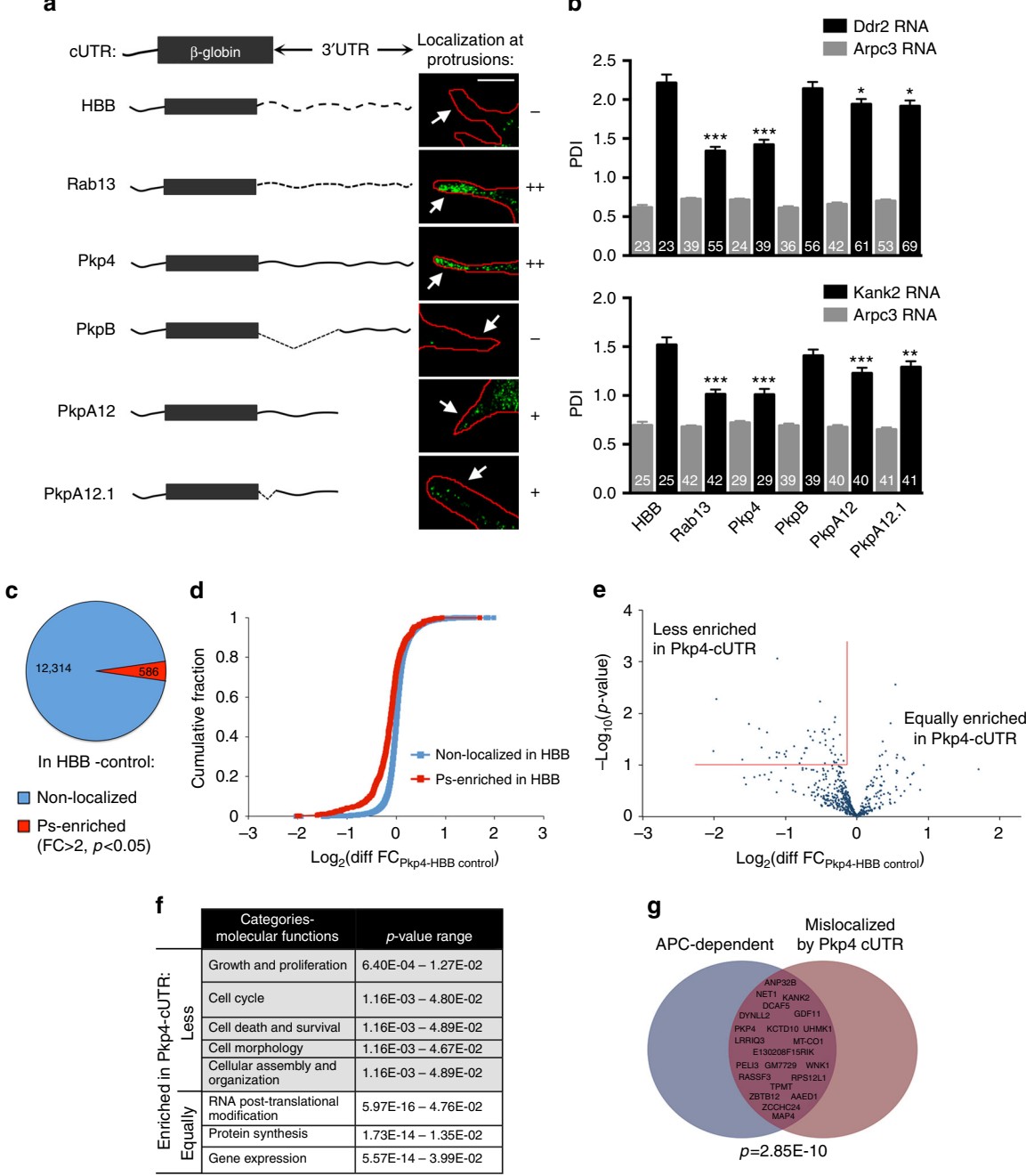

**Fig. 7** A competition-based approach preferentially mislocalizes APC-dependent RNAs from protrusions. **a** Schematic of exogenously expressed constructs containing the β-globin coding sequence followed by the indicated 3′UTRs (β-globin UTR (HBB; spaced dashed line); Rab13 UTR (dashed line), Pkp4 UTR (solid line), or deletion fragments of Pkp4 UTR (PkpB, PkpA12, and PkpA12.1)). Representative in situ hybridization images detecting the β-globin RNA, and the ability of each construct to localize at protrusions are shown. Images focus on individual protrusions. Scale bar: 10 μm. Whole-cell images and PDI values are shown in Supplementary Fig. 7a, b. **b** Cell lines, stably expressing β-globin constructs with the indicated UTRs, were analyzed by in situ hybridization and RNA distributions were assessed through PDI quantitation. (Ddr2, Kank2: APC-dependent RNAs; Arpc3: control, diffuse RNA). *N*-values of observed cells, from at least 3 independent experiments, are indicated within each bar. Error bars: standard error. *p*-values: ***<0.001, **<0.01, *<0.05 by analysis of variance with Dunnett's multiple comparisons test. **c** Pie chart of non-localized or Ps-enriched RNAs in control HBB cells from 4 replicate experiments. Cutoffs for Ps-enrichment were set at FC > 2 and *p*-value < 0.05. **d** Cumulative fraction plot of log2 fold-change differences between control HBB and Pkp4 cUTR-expressing cells, for RNAs non-localized or localized in protrusions of control HBB cells. **e** For all Ps-localized RNAs of control HBB cells, the log₂ differences in FC values, between HBB control and Pkp4 cUTR-expressing cells, were plotted against the corresponding *p*-values. Applying the cutoffs marked by the red lines (see Supplementary Data 3 for details), Ps-localized RNAs were distinguished into groups less- or equally-enriched in protrusions upon Pkp4 cUTR expression. **f** Categories of molecular functions, derived through IPA analysis, significantly represented in RNA groups equally- or less-enriched in protrusions of Pkp4 cUTR-expressing cells. **g** Results of differential enrichment analysis showing significance of overlap between APC-dependent RNAs and RNAs mislocalized upon Pkp4 cUTR expression

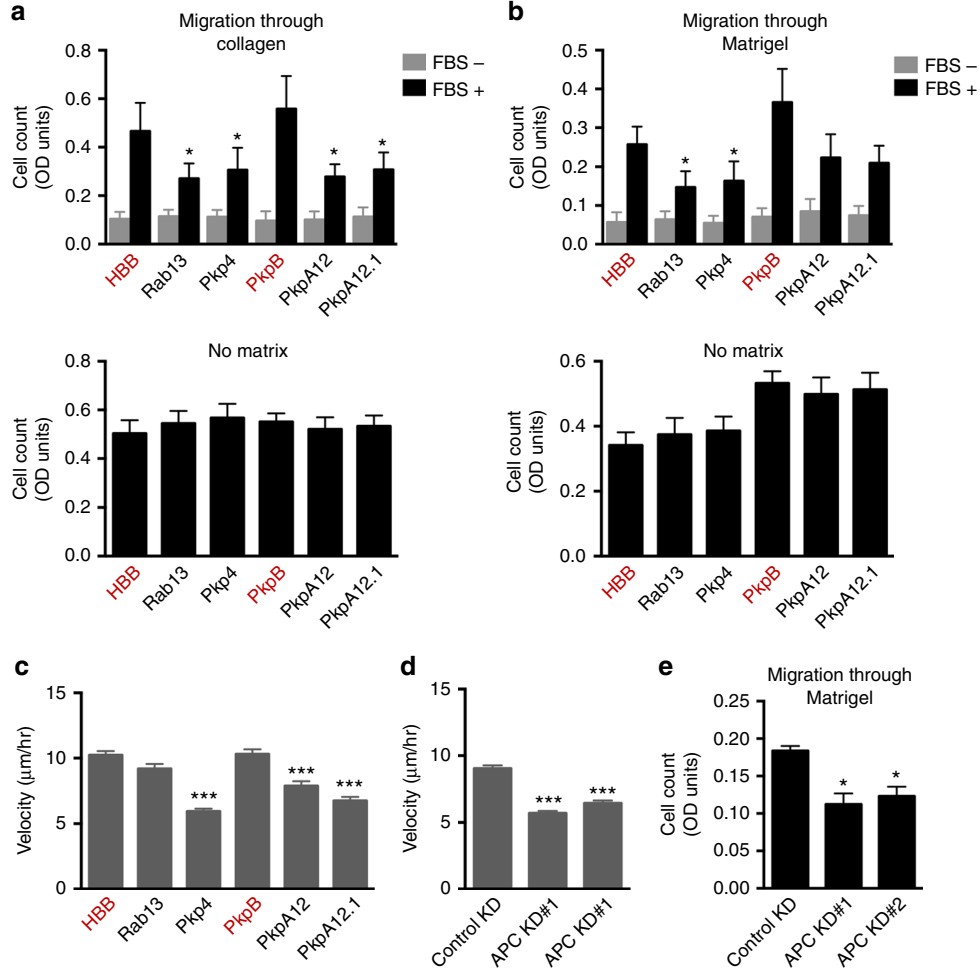

**Fig. 8** Localization of APC-dependent RNAs is required for efficient migration on 2D and 3D substrates. **a, b** Cell lines, stably expressing β-globin constructs with the indicated UTRs, were plated on top of collagen gel **a** or Matrigel **b** and were induced to migrate by addition of serum (FBS) in the bottom chamber. The amount of cells reaching the bottom was quantified from at least 5 independent experiments, (error bars: standard error). Lower panels show the amount of cells reaching the bottom in the absence of matrix, in parallel experiments, to control for cell viability during the assay. Note that because of the length of the assay, migration in the absence of matrix does not reflect migration speed. **c, d** Cells stably expressing constructs with the indicated UTRs **c** or transfected with the indicated siRNAs **d**, were monitored while randomly migrating. Average velocity values were derived from multiple individual cell tracks in two independent experiments observing 40–70 cells in each. **e** Migration through Matrigel of control or APC knockdown cells. $p$-values: ***< 0.0001, *< 0.05 against HBB or si-Control, by analysis of variance with Bonferroni's multiple comparisons test

and cell bodies from the control HBB and the Pkp4 cUTR cells. RNAs enriched at protrusions were globally identified through RNA-Seq, as described above (Fig. 1). Principal component analysis showed that replicates clustered together, indicating good reproducibility (Supplementary Fig. 8). In agreement with our previous observations, in the control HBB cells, ca. 4.5% of detected RNAs (586 of 12,900 RNAs) were significantly enriched in protrusions (FC > 2, $p$-value < 0.05, see Methods section) (Fig. 7c and Supplementary Data 3). To determine how enrichment of these RNAs was affected by Pkp4 cUTR expression, we assessed the $\log_2$ FC difference between HBB control and Pkp4 cUTR cells, $\log_2(\text{Diff FC}) = \log_2(\text{FC}_{\text{Pkp4}}) - \log_2(\text{FC}_{\text{HBB control}})$. Positive values indicate increased enrichment, while negative values indicate decreased enrichment upon Pkp4 cUTR expression. Similar to what was observed in the case of APC knockdown (Fig. 1b), a subset of Ps-localized RNAs became less enriched upon Pkp4 cUTR expression shifting the cumulative fraction plot to negative values (Fig. 7d, red line), while the remaining Ps-localized RNAs were not affected, with $\log_2$Diff FC values centering around zero (Fig. 7d, red line). By contrast, $\log_2$Diff FC values of non-localized RNAs predominantly

centered around zero (Fig. 7d, blue line), indicating that Pkp4 cUTR expression does not result in considerable overall changes in RNA distributions. These results indicate that, similar to what was observed upon APC knockdown, distinct RNA subsets are differentially affected by Pkp4 cUTR expression for their localization at protrusions.

Similar stringency cutoffs as before, based on $\log_2(\text{Diff FC})$ and $p$-values, were used to define Ps-localized RNAs that become less enriched upon Pkp4 cUTR expression in contrast to those that remain equally enriched (48 and 527 RNAs were contained in each category, respectively) (Fig. 7e, red lines and detailed in Supplementary Data 3). IPA analysis showed that the affected RNAs belonged to similar functional categories as the APC-dependent RNAs (i.e. cellular growth, morphology, assembly, and organization; Fig. 7f and Supplementary Data 2; compare with Fig. 1d). By contrast, "protein synthesis" and "gene expression" were two of the highest represented functional classes among RNAs whose enrichment was not affected by cUTR expression, similar to the APC-independent RNAs described above.

Importantly, 34% of RNAs mislocalized by Pkp4 cUTR expression overlap with APC-dependent RNAs. To determine

the statistical significance of this overlap, we performed gene set enrichment analysis. This analysis revealed that the group of RNAs affected by Pkp4 cUTR expression is highly similar to APC-dependent RNAs, with an adjusted *p*-value of 2.85E-10 (Fig. 7g) (Methods section). We note that the number of identified protrusion-enriched RNAs from samples sequenced independently can vary, likely due to technical reasons (e.g., compare Figs. 1a and 7c). Given this variability, the fact that we observe only a partial overlap is not unexpected, even though we cannot rule out the existence of independently regulated RNAs. Nonetheless, these data indicate that cUTR overexpression, to a significant extent, mislocalizes APC-dependent RNAs from protrusions.

**APC-dependent RNAs are required for 2D and 3D migration.** We used this system to address the functional role of APC-dependent RNA localization. The Ddr2 protein, encoded by an APC-dependent RNA, has been previously implicated in cell invasion[39, 40], we thus measured the ability of cells to migrate in response to a high concentration of serum through a 3-dimensional Collagen I or Matrigel matrix. Interestingly, expression of both Rab13 and Pkp4 cUTR constructs significantly reduced the ability of cells to migrate through 3D matrices (Fig. 8a, b). These differences were not caused by changes in cell proliferation (Supplementary Fig. 9a) and in the absence of matrix a similar number of cells reached the bottom of the insert (Fig. 8a, b; bottom panels). Importantly, testing of the further Pkp4 deletion fragments showed that reduced migration was caused by overexpression of the localization-competent fragments (PkpA12 and PkpA12.1), but not by the non-localized PkpB fragment (Fig. 8a, b), consistent with the idea that the effects are mediated through sequestration of a localization-related factor.

To assess more precisely what migration parameters are affected, we assessed migration speed and directionality of cells randomly moving on 2-dimensional fibronectin-coated surfaces. Expression of cUTR constructs did not affect the directionality of migration (Supplementary Fig. 9b), but significantly affected migration speed (Fig. 8c). Intriguingly, in this case, the effect caused by overexpression of the Rab13 UTR was not as pronounced as that of the Pkp4 UTR or its deletion fragments, suggesting that distinct UTRs might affect overlapping RNA sets that are differentially required for 2D and/or 3D migration. We acknowledge that without knowing the identity of the factor(s) being titrated, the above data cannot unequivocally prove, but we believe they strongly indicate that APC-dependent RNA mislocalization (rather than titration of factors linked to other processes) underlies the observed migration defects.

To further support this idea, we assessed the migration of APC knockdown cells. Indeed, upon delivery of two different siRNAs against APC, we observed a decrease in migration velocity similar in magnitude to that observed in the case of cUTR-expressing cells (Fig. 8d). Migration defects upon APC loss have been previously reported[41–43], however since loss of APC results in significant activation of Wnt-signaling, and of β-catenin transcriptional activity, it has been difficult to determine to what extent the role of APC in migration is mediated through transcriptional effects of activated β-catenin vs. other APC functions that are independent of β-catenin activation. Indeed, as expected, siAPC-treated cells show a significant increase in baseline Wnt-signaling activation (i.e., in the absence of Wnt ligand), however, importantly, cUTR-expressing cells do not show any appreciable Wnt-signaling activation (Supplementary Fig. 9c). Therefore, cUTR expression affects the localization of APC-dependent RNAs without affecting other APC functions. Overall, these data support the model that localization of RNAs through APC directly contributes to cell migration.

## Discussion

Cells modulate their contractility by sensing the stiffness of the extracellular environment and initiating a series of mechanotransduction events, which largely converge on activation of the RhoA GTPase[25, 44]. RhoA effectors such as the kinase ROCK and the formin mDia subsequently affect the properties and dynamics of both the actin and microtubule cytoskeleton[44–46]. The work reported here reveals that a functionally important consequence of these signaling events is the localization of APC-dependent RNAs at cellular protrusions. On stiff substrates, increased actomyosin contractility and Rho GTPase signaling promote formation of a network of post-translationally modified, detyrosinated microtubules, which is required to support APC-dependent RNA localization.

We provide evidence for the functional importance of this RNA localization pathway by using a competition-based approach to interfere with APC-dependent RNA localization. Importantly, this results in defects in cell migration in 2D and 3D systems. We speculate that localization of RNAs at protrusive regions in response to increased tension might underlie the increased cell migration and invasive behavior promoted upon generation of tension through contraction of the actomyosin cytoskeleton[24].

Localized RNAs often exert their functions through directing local protein synthesis. Indeed, local RNA translation within protrusions is required for their stabilization[11]. Furthermore, APC-dependent RNAs are involved in pathways functionally relevant to cell movement, such as cell morphology, signaling and organization (Fig. 1d), and encode proteins (e.g., Ddr2 and Rab13) with known roles in migration and invasion[39, 47]. We, therefore, favor the hypothesis that local translation of APC-dependent RNAs accounts for their contribution to cell migration.

Interestingly, we also show that additional mechanisms direct RNAs at protrusions in a manner that is independent of APC. Such APC-independent RNAs comprise a group with distinct functionalities and preferential localization to different types of protrusions than the APC-dependent ones. This raises the possibility that the specific cytoskeleton and signaling characteristics of protrusions direct their RNA content and eventual functional contributions. It would be interesting to determine whether the degree of actomyosin contractility in different types of protrusions determines the balance of MT modifications, as discussed below, and thus directs transport of distinct RNA subsets.

Prominent among the APC-independent RNAs are RNAs encoding r-proteins. R-protein mRNAs have been observed in a number of screens for localized RNAs in various cell types[13, 21, 48, 49]. Through comprehensively detecting all r-protein transcripts, we show here that virtually all r-protein mRNAs are localized at protrusions and that the degree of their localization is coordinately regulated. We believe that this renders unlikely the possibility that these local r-protein transcripts serve extra-ribosomal functions. Their coordinated regulation is consistent with a ribosome-related function and supports proposed hypotheses such as de novo ribosome assembly at peripheral sites or ribosome repair[48, 49]. A better understanding of the regulatory signals and the physiological contexts that promote the localization of r-protein mRNAs would provide useful insights into deciphering their functional roles.

Another important aspect revealed by this work concerns the role of modified MTs in RNA transport. Microtubules and microtubule-based motors are well known to be required for transport of RNAs in various systems[50, 51]. However, cellular microtubules exhibit a large degree of heterogeneity stemming from the accumulation of various post-translational modifications

on the tubulin subunits, such as detyrosination and acetylation. These microtubule modifications are thought to constitute a microtubule code and affect microtubule recognition by kinesin motors and thus cargo transport and microtubule functions[52, 53]. How MT modifications impact transport of RNA cargoes is largely unexplored. To our knowledge, APC-dependent RNAs are the first group of RNAs that require a specific set of modified microtubules, namely detyrosinated microtubules, for their localization, as shown in this work and supported by additional recent work from our group[54].

Interestingly, actomyosin contractility appears to exert distinct effects on modified MTs. As we show here, increased contractility enhances MT detyrosination through the RhoA effector mDia1. By contrast, increased contractility leads to a decrease in acetylated MTs[55, 56] through a mechanism involving the action of the myosin phosphatase MYPT1 on the MT deacetylase HDAC6[57]. This suggests that the levels of individual modifications can change independently of the others, consistent with prior reports[58]. Intriguingly, loss of acetylated MTs leads to activation of RhoA through the Rho exchange factor GEF-H1[55] raising the possibility that RhoA is part of a feedback mechanism that coordinates the levels of modified MTs. The level of actomyosin contractility and RhoA activation could therefore modulate the efficiency and/or direction of transport of RNAs or other cargo.

Our results describe a mechanism whereby changes of the actomyosin cytoskeleton affect MT-based RNA localization at protrusions. Interestingly, evidence indicates that a similar interplay between actin and MT-based processes could operate in the case of other RNA targets. Apart from APC-dependent RNAs, the RNAs encoding β-actin and subunits of the Arp2/3 complex are localized in lamellipodia. These RNAs are not significantly localized in our system, likely because of their dependence on the ZBP1 protein, which is more highly expressed in embryonic and cancer cells[8, 17]. Nevertheless, while the β-actin RNA associates with MTs and requires a kinesin motor for its transport[59, 60], its localization is affected by the actin-based motor MyoVa and RhoA signaling[23, 61]. MyoVa also affects the distribution and trafficking properties of other RNA-binding proteins (i.e., FMRP and FUS) that transport RNAs along microtubules[62, 63]. The underlying mechanisms are unclear, however, it is interesting that MyoV apart from its role in cargo transport can additionally affect the organization of actin cables through modulating their dynamics and tension[64]. It might be interesting to speculate that, in a manner analogous to the one described here, changes in the mechanical properties of the actin cytoskeleton through MyoV action could impact MT-based RNA transport by altering MT dynamics or modifications.

## Methods

**Cell culture and protrusion/cell body isolation.** NIH/3T3 mouse fibroblast cells (ATCC) were grown in DMEM (Dulbecco's Modified Eagle Medium) supplemented with 10% calf serum, sodium pyruvate, and penicillin/streptomycin (Invitrogen) at 37 °C, 5% $CO_2$. Cell line has not been authenticated. It has been tested yearly for mycoplasma and is free of contamination. Protrusions and cell bodies were isolated from serum-starved cells plated for 2 h on Transwell inserts equipped with 3.0-μm or 0.4-μm porous polycarbonate membrane (Corning) as previously described[13, 32]. Briefly, 1.5 million cells were plated per 25mm filter and 1 or 3 filters were used for cell body or protrusion isolation, respectively. LPA was added to the bottom chamber for 1 h and the cells were fixed with 0.3% paraformaldehyde for 10 min. For isolating protrusions, cell bodies on the upper surface were manually removed by wiping with cotton swab and laboratory paper. The protrusions on the underside (or, in the case of the 0.4-μm membranes, the protrusions remaining within the pores) were then solubilized by immersing the filter in crosslink reversal buffer (100 mM Tris pH 6.8, 5 mM EDTA, 10 mM DTT, and 1% SDS) and gentle scraping. Cell bodies were similarly isolated after manually removing protrusions from the underside of the membrane. The extracts were incubated

at 70 °C for 45 min and used for protein analysis or RNA isolation using Trizol LS (Invitrogen).

**siRNA transfection and generation of stable lines.** Stable cell lines were generated by lentiviral transduction and selected with 800 ng ml$^{-1}$ geneticin (Thermo Fisher Scientific Cat No. 10301). Exogenous gene expression was induced with 1 μg ml$^{-1}$ doxycycline for 72 h.

NIH/3T3 cells were transfected with siRNAs to a 40 pmol final concentration with Lipofectamine RNAiMAX (Thermo Fisher Scientific, cat# 13778-150) according to the manufacturer's instructions. Cells were analyzed 72 h after transfection.

siRNAs used were as follows (Qiagen):
Myh9 #1: (cat # SI01321411, target sequence: 5′-CAGGGCTTATCTACACCTATT-3′); Myh9 #2: (cat# SI01321418, target sequence: 5′- TCCAGCAAGAATGGCTTTGAA-3′); Myh9 #3: (cat# SI01321425, target sequence: 5′-CTCGAGAAAGTCCACTCGGAA-3′); Myh9 #4: (cat# SI01321432, target sequence: 5′-AGAGGAGATCGTGGAGATGTA-3′); TTL #1: (cat# SI01458345, target sequence: 5′- CCGCAAAGCTTCCTTAGTAAA-3′)
TTL #3: (cat# SI01458331, target sequence: 5′-CACCGCAAGTTTGACATTCGA-3′);
APC: (cat# SI00900186, target sequence: 5′-AAGGAAGTACTTAAGCAGCTA-3′ and cat# SI00900179, target sequence: 5′-AAGGACTGGTATTATGCTCAA-3′);
AllStars Negative control siRNA (cat# 1027281).

**Plasmid constructs and lentivirus production.** To express *β-globin* RNA followed by different 3′UTRs, the genomic region containing the coding sequence of human β-globin was ligated to fragments corresponding to the following UTRs: HBB (3′UTR of human β-globin, Acc#: NM_000518.4), Rab13 (3′UTR of mouse Rab13, Acc#: NM_026677.4), Pkp4 (3′UTR of mouse Pkp4, Acc#: NM_026361.2), PkpB (nts 394-756 of mouse Pkp4 3′UTR), PkpA12 (nts 1-297 of Pkp4 3′UTR), and PkpA12.1 (nts 80-297 of mouse Pkp4 3′UTR). The sequences were cloned into the pENTR1A vector and then transferred into the pINDUCER20 lentivector (Addgene) using the Gateway LR clonase II Enzyme mix (Thermo Fisher Scientific, cat# 11791-020). Lentivirus was produced in HEK293T cells cultured in DMEM containing 10% FBS and Penicillin/Streptomycin. HEK293T cells were transfected with pINDUCER20 lentivectors, together with packaging plasmids pMD2.G and psPAX2 using PolyJet In Vitro DNA transfection Reagent (SignaGen) for 48 h. Harvested virus was precipitated with Polyethylene Glycol at 4 °C overnight.

**Drug treatments.** Cells were treated with 50 μM Blebbistatin (Sigma) for 2 h during spreading or 10 μM Y27632 for 1 h (Sigma). For parthenolide treatment, cells were allowed to spread for 2 h before being treated with 5 μM parthenolide for another 2 h (Sigma-Aldrich, cat# P0667). In the case of LPA, cells were serum-starved overnight, allowed to attach to gels for 2 h and 10 μM LPA was added for 2 h. In the case of Rho Activator II (Cytoskeleton, Inc.), the cells were serum-starved by changing media to 1% calf serum for 24 h and then to no serum overnight. The cells were then plated on 1 kPa gels for 2 h and Rho Activator II was added at 0.25 μg ml$^{-1}$ for 2 h.

**RNA analysis.** For RNA-Seq analysis, ca. 1 μg of total RNA from 4 replicates of protrusion and cell body samples was used to construct stranded, total RNA libraries using TruSeq Stranded mRNA Library Prep kit (Illumina RS-122-2201). Amplified libraries were sequenced on an Illumina HiSeq2500 sequencer using Illumina TruSeq V4.0 chemistry and paired-end, 126-bp reads. The samples yielded 35–44 million mapped reads (for the control and APC knockdown samples) and 50–77 million mapped reads (for the HBB control and Pkp4 cUTR-expressing samples). The raw RNA-Seq fastq reads were aligned to mouse genome (mm10) using STAR (v.2.4.2a) on 2-pass mode with mouse gencode (release 4) gtf. Genes were subsequently counted using Rsubread (v. 1.20.1), normalized using limma-voom (limma, v. 3.26.1)[65, 66] and applying log-counts per million (log-cpm) > 0.5 in at least one sample as an expression cutoff. Statistical tests on gene expression changes were performed using the empirical Bayes moderated $t$-test as implemented in the limma Bioconductor package[67].

Gene set enrichment analysis using a hypergeometric distribution analysis was performed on gene sets downloaded from various databases, including Gene Ontology Consortium, KEGG, and Molecular Signatures Database (http://software.broadinstitute.org/gsea/msigdb/index.jsp), Pathway Commons (http://www.pathwaycommons.org/pc2/datasources), and Panther (http://pantherdb.org/data/) resulting in 16,564 gene sets. A custom gene set, comprised of the APC-dependent RNAs, was added. A hypergeometric test was used to calculate the enrichment of genes among the gene sets, followed by a false discovery rate adjustment using the Benjamini–Hochberg procedure.

For nanoString analysis, total RNA from protrusion and cell body samples was analyzed using custom made codesets and the nCounter analysis system according to the manufacturer's instructions. Probes detecting r-protein mRNAs were designed, when possible, to distinguish between transcripts derived from r-protein genes and pseudogenes.

For real-time RT-PCR analysis, total RNA was reverse transcribed using the iScript cDNA synthesis kit (Bio-Rad) and PCR was performed using Power SYBR Green PCR master mix (ThermoScientific) on a 7900HT Fast real-time PCR system (Applied Biosystems).

For fluorescence in situ hybridization (FISH), cells were plated on fibronectin-coated polyacrylamide gels or fibronectin-coated glass coverslips for 2–3 h and subsequently fixed in 4% paraformaldehyde (PFA) for 10 min. FISH was performed with QuantiGene ViewRNA ISH Cell Assay kit (Affymetrix, cat# QVCM0001) according to the manufacturer's instructions. The Affymetrix probe sets used were: Ddr2 cat# VB1-14375, Kank2 cat# VB1-14376, Pkp4 cat# VB1-14506, RhoA cat# VB6-14572, Cyb5r3 VB1-18647, and Arpc3 cat# VB6-14571. To detect PolyA RNAs, LNA modified oligodT probes (30 nucleotides) labeled with ATTO-655 were added during hybridization, pre-amplification, amplification, and last hybridization steps of QuantiGene ViewRNA ISH Cell Assay. Nuclei were stained using DAPI (4′,6-diamidino-2-phenylindole) and cell outline was obtained by staining either for 18S ribosomal RNA or using Cell mask stain (Thermo Fisher Scientific).

**Preparation of polyacrylamide gel substrates**. Thin polyacrylamide gels were prepared on glass coverslips as described previously[68]. Concentrations in this work included 15% acrylamide + 1.2% bis acrylamide (280 kPa), 8% acrylamide + 0.07% bis acrylamide (5 kPa), and 3% acrylamide + 0.2% bis acrylamide (1 kPa). Gels were coated with 0.1 mg ml$^{-1}$ fibronectin (Sigma-Aldrich) overnight. Young moduli of the gels were determined using atomic force microscopy and dynamic mechanical analysis. Characterization of the surface-bound fibronectin was done using immunofluorescence (IF) to ensure equal protein concentration with stiffness.

**Immunofluorescence and western blot**. Antibodies used were the following: anti-Glu-tubulin (Abcam, cat# ab48389, 1/250 dilution for IF, and 1/1000 for WB); anti-α-tubulin (Sigma-Aldrich, cat# T6199, 1/1000 dilution); anti-actin (Ab-5, BD Biosciences, cat# 612656, 1/4000 dilution); anti-FAK (BD Biosciences, cat# 610088, 1/250 dilution); anti-pFAK(Y397) (Cell Signaling Technology, cat# 3283 S, 1/250 dilution); anti-phospho-myosin light chain 2 (Ser 19) (Cell Signaling Technology, cat# 3671, 1/200 dilution); anti-myosin (clone MY-21, Sigma cat# M4401, 1/250 dilution); anti-TTL (Proteintech cat# 13618-AP, 1/500 dilution). Images of uncropped WBs are shown in Supplementary Fig. 10.

For IF, cells were plated onto gels and were fixed with 4%PFA in PBS (phosphate-buffered saline) for 15 min, permeabilized with 0.2% Triton X-100 in PBS for 5 min, blocked with 3% BSA for 1 h, and incubated with primary antibodies overnight at 4 °C. Secondary antibodies were conjugated with Alexa 488, 546, or 647 (Alexa Fluora series from Thermo Fisher Scientific). Nuclei or whole-cell area was stained with DAPI or cell mask blue stain, respectively. For determination of percentage of cells with a Glu-MT network, positive cells were defined as those having > 10–15 distinctly stained Glu-MT fibers.

**Image acquisition and PDI analysis**. IF images were obtained using a AMG EVOS fl microscope (equipped with a 40x Plan Coverslip Corrected FL Air objective, 0.75NA) or a Leica confocal SP8 microscope (equipped with a HC PL APO 63x oil CS2 objective). RNA FISH images were acquired with a Leica confocal SP8 microscope. Z-stacks through the cell volume were obtained and maximum intensity projections were used for subsequent analysis. For PDI analysis, Cell mask stain (Thermo Fisher Scientific) was used to obtain a binary cell mask of the whole-cell area. DAPI stain was used to obtain a binary mask of the nuclear area. RNA FISH images were background subtracted. Signal within the nuclear area was also subtracted when detecting polyA RNA. The processed images were then analyzed using a MATLAB script described in ref. [33] and modified to calculate a PDI by measuring the second moment of RNA pixel intensity positions relative to the centroid of the nucleus.

We note that there are some considerations that limit the ability to directly compare PDIs across cell populations, when the cells exhibit significantly different spreading areas. First, the subtraction of the nuclear signal causes an increase in PDI (PDI = 1 if the signal is completely diffuse throughout the cell; when subtracting the central nuclear signal, a completely diffuse cytoplasmic signal shows a PDI > 1). This increase is disproportionate between cells with very different spreading areas. That is because in cells with smaller spreading area (such as for those on soft substrates) a relatively larger proportion of the cellular area is covered by the nucleus. While this does not affect the comparison of PDI values of different RNAs within the same cell, it nevertheless prevents a direct comparison to cells with larger spreading areas, where the nuclear signal covers a smaller proportion of the overall cellular area. Second, cells with small spreading area tend to be rounder with uniform distribution of the cytoplasmic volume across the cell body. By contrast, when cells have a large spreading area, protrusive regions are much thinner with most of the cytoplasm being present around the nucleus. This is reflected in a change in the distribution of polyA RNA, which becomes more perinuclear as the spreading area of the cell increases. We, thus, do not think it is valid to compare directly PDI values between cells with very different spreading areas. Comparisons with internal control RNAs show whether an RNA exhibits a preferential localization. PDIs can be compared across different conditions only when treatments do not affect significantly the cellular area, which

can also be seen by the lack of change in the corresponding polyA RNA PDI values. We routinely present all individual PDI values to depict the underlying variability within cell populations.

**Traction force microscopy**. Cell traction forces were measured by embedding 0.2 μm fluorescent beads in the upper surface of polyacrylamide gels. Images were taken in the "stressed" state with the cell attached to the substrate. The cell was subsequently detached using trypsin and another image was taken in the "null" state. To correct for experimental drift, the imageJ plugin "align slices in stack" was used. The displacement field was subsequently calculated using particle image velocimetry (PIV), as previously described[69]. With the displacement field obtained through the PIV analysis, the traction force field was reconstructed by Fourier traction cytometry. The PIV and traction force microscopy macros are available online and can be implemented in ImageJ.

**Migration assays**. Random cell migration was monitored with Vivaview incubator microscope system (Olympus Corp.). Cells were plated on 1 μg ml$^{-1}$ fibronectin-coated glass bottom microwell dishes (MatTek Corp. Cat No. P35G-1.5-14-C) and equilibrated in the incubator for 2 h. Live cell time-lapse images were taken at 20-min intervals for 24 h. Cell movements were tracked using manual tracking tool from ImageJ. Movement parameters (velocity and directionality) were obtained using chemotaxis tool form ImageJ.

Transwell invasion assay was performed based on ref. [70] with some modifications. 24-well plates with 8 μm pore size inserts were used. Rat-tail collagen type I (BD Biosciences, cat# 354236) was prepared at 0.3 mg ml$^{-1}$ and Matrigel (BD Biosciences, cat# 354324) was used at 20% dilution. 70 μl of matrix solution was added to the inserts and let to solidify. 50,000 cells were plated onto each insert in 250 μl serum-free medium. 10% FBS was added in the lower chamber to induce cell motility. 48 h later the membrane was fixed in 70% ethanol, cells and gel material were removed from the upper chamber and the membrane was stained with 0.5% crystal violet. The absorbed dye was eluted in 30% acetic acid and cell count was measured by optical densiity at 600 nm on a plate spectrophotometer.

**TOPflash luciferase assay**. To assess Wnt-signaling and β–catenin transcriptional activity, cells were transfected with TOPFlash reporter and pRL-TK plasmids, using Lipofectamine 2000 (Invitrogen). Firefly and Renilla luciferase activities were measured using the dual-luciferase reporter assay system (Promega Corporation, Wisconsin), according to the manufacturer's instructions, using a Veritas microplate luminometer (Turner Biosystems). Firefly luciferase levels were normalized to those of Renilla luciferase for each sample.

**Statistics**. Prism 7 by GraphPad software was used for graph generation and statistical analyses. For multiple comparisons, one-way analysis of variance (ANOVA) was used with Bonferroni's or Dunnett's multiple comparisons test. Two-way ANOVA was used when there were more than two variables. For comparison of two data sets, Mann–Whitney test was used. Significance level was set to $p < 0.05$. Number of independent biological replicates and sample sizes analyzed are stated in the figure legends.

**Data availability**. The data that support the findings of this study are available from the corresponding author upon reasonable request.

MATLAB script for PDI calculation is available upon request.

RNA-Seq data have been submitted to GEO (Accession GSE82244).

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

## Acknowledgements

We would like to thank Parthav Jailwala of the CCR Collaborative Bioinformatics Resource (CCBR) for help with analysis of RNA-Seq data, and the CCR Genomics Core for support. This work was supported by the Intramural Research Program of the Center for Cancer Research, NCI, National Institutes of Health (S.M.) and the Human Frontiers Science Program RGP0058/2012 (H.A.-E.). We would like to acknowledge funding support from the NCI-UMD Partnership for Integrative Cancer Research (S.M. and H.A.-E.).

## Author contributions

T.W. and S.H. designed, performed, and analyzed the experiments. M.C. analyzed RNA-Seq data. H.A.-E. designed the experiments. S.M. designed, performed, and analyzed experiments, and wrote the manuscript.

## Additional information

**Competing interests:** The authors declare no competing financial interests.

