## [Peer Review File · Nature Communications]

Reviewers' Comments:

Reviewer #1:

Remarks to the Author:

Summary.

mRNAs are actively transported to many locations in diverse cell types and this process is evolutionarily conserved. Previous work from this group has demonstrated that a distinct set of mRNAs are localized to cell protrusions and that the localized RNAs are associated with the APC protein and Glu-microtubules. The current work expands on the previous studies to provide a genome-wide view of all the RNAs that require the APC protein for localization to protrusions. Additionally, the authors show that cell contractility influences the efficiency of mRNA localization through a pathway that regulates Glu-microtubule formation, although they are not able to establish a molecular link between these two pathways. Finally, the authors use an overexpression and competition experiment to show that mislocalization of a subset of APC-dependent RNAs results in defects in cell motility in 3D media. Ascribing a function for APC-dependent mRNA localization is a potentially very important addition to the literature. While the authors have provided a suggestion that mRNA localization is important for cell motility I believe that a few additional experiments are required to clearly demonstrate this important point. If the authors can address this issue and a one additional minor concern I would enthusiastically support publication and believe that this manuscript will be a valuable addition to the literature.

Major Points:

1. In Figure 8 the authors show that cells overexpressing Pkp4, Rab13, PkpA12, and PkpA12.1 cUTRs show reduced motility through 3D substrates. The magnitude of the motility effects (Figure 8) and effects on mRNA localization (Figure 7) are different for each of the constructs. The authors have not carefully characterized all aspects of each of these cUTR constructs. From the images provided in Figure 7 and Figure S7 it is clear that these constructs do not localize to protrusions with equal efficiency. I would like to see that authors quantitate localization efficiency (using PDI) for each of the cUTR constructs used in Figures 7 and 8.
2. In Figure 7 the authors show that overexpression of Pkp4 cUTR results in the mislocalization of a substantial fraction of APC-dependent mRNAs. However, there are many mRNAs that show increased localization to protrusions and many APC-dependent mRNAs that are not mislocalized. In Figure 8 the authors show that overexpression of cUTR constructs results in motility defects, linking mRNA localization to cell motility, which is a very exciting finding. However, because the cUTR experiment clearly affects many mRNAs that are not regulated by APC it is important to have an orthogonal method to show that APC-dependent mRNA localization affects cell motility to confirm this finding. In Figure 1 the authors use APC siRNA to define APC-dependent mRNA localization. It would solidify the link between mRNA localization and cell motility presented in Figure 8 to look at cell motility in APC siRNA treated cells.

Minor points.

1. For each of the Figures showing PDI data the authors present a plot with points, mean and SE. In the Figure legends that authors state that each of the points is an individual cell and that the experiments have been repeated at least twice. I assume that that mean value is the mean of the pooled cells rather than the mean of the replicate experiments. I think that it would be more useful to present that mean value of the replicate experiments and the errors as the SD or SEM of the variation between biological replicates. The N of cells measured could then be stated in the legend. This presentation would give a more accurate depiction of the reproducibility of the experiment.

Reviewer #2:

Remarks to the Author:

This well-written manuscript provides evidence that the stiffness of extracellular matrix and

cellular contractility can both control the localization of specific mRNAs to cellular protrusions, and that an mRNA subclass can be implicated in promoting cell migration using a novel cUTR competition approach that can mislocalize endogenous APC-dependent mRNAs. On the positive side, this manuscript provides new evidence linking extracellular matrix stiffness, cellular traction force, and actomyosin contractility to the specific localization of subclasses of mRNAs, as well as providing some insight into the biological significance of this phenomenon of localization to cellular protrusions; in terms of methods, the use of four independent replicates for RNA-seq is excellent, and the manipulations of Glu-microtubule levels are convincing. Apparent negative elements, however, include issues involving the interpretation of some of the localization data, uncertainty about the strength of some of the conclusions, and problems with statistical analyses. Many of the specific concerns may be possible to resolve by a reconsideration of some interpretations and rewording; a couple of the conclusions, however, may need revision. Nevertheless, this manuscript provides an intriguing advance in a research area that needs more of this type of exploration of the roles of local mRNA localization, its control, and its physiological importance. Consequently, even though a number of points are listed below, the intent is to strengthen this potentially valuable paper rather than to reject it.

1. There appears to be a basic conceptual problem in the interpretation of altered localization as quantified by PDI when polyA RNA PDI is used as the baseline for comparison, which occurs in Figures 3b and 3h. The interpretation becomes inconsistent with direct comparisons between Ddr2 RNA levels because there is sometimes an anomalous substantial decrease in polyA RNA PDI. Puzzlingly, such seemingly more-appropriate direct comparisons without this form of normalization are shown elsewhere, e.g., in Figure 3d and Figure 6b and 6d.

To be specific, Figure 3b would probably show no change in quantified Ddr2 RNA on substrates of different stiffness if these values were compared with each other. Instead, the authors perform comparisons with polyA RNA, which clearly moves centrally in cells. The biological or molecular significance of this latter movement is not clear, which should also be discussed. But the key point of this paper involves the localization of the APC-dependent subclass of RNAs, which clearly do not appear to change with stiffness. Strangely, the authors do perform this appropriate comparison in Figure 3d, where inhibition of contractility results in a significant alteration in the distribution of Ddr2 RNA. There appears to be a fundamental disconnect between sections b and d of this figure. Similarly puzzling is the failure to compare alterations in PDI consistently in Figures 3f versus 3h, where the effect of LPA may still be statistically significant, but perhaps not so clear with a Rho Activator II; these comparisons using more appropriate statistical analyses need to be performed [point #3], rather than trying to compare with polyA RNA, which is decreased by these Rho activators.

2. A related concern is that the wrong comparisons in Figure 5 and Figure 6 were performed in terms of statistical analyses, even though this reviewer thinks that the results might potentially help to support the authors' interpretations. Specifically, in Figure 5b, the correct comparison is performed, yet the same comparison is missing from Figure 5c, in which the comparison is suddenly with polyA RNA, which is an inconsistent analysis compared with the preceding figure. In Figure 6b, the most important analysis needs to be between the si-Control values and the si-TTL values, where it appears that there will be the hoped-for statistically significant rescue of the +blebbistatin condition by si-TTL, at least at a statistically significant level. Similarly, the rescue by mDia in the presence of blebbistatin is not compared.

3. An important problem occurs in multiple important figures with respect to the choice of the method of statistical analysis (which is likely to result in apparent levels of statistical significance that are actually not correct) by using Student's t-test for experiments in which there are multiple variables. One cannot simply choose a pair of variables to compare without applying an appropriate statistical correction to the t-test, such as the Bonferroni correction. The alternative is to perform ANOVA with a statistically appropriate post-test. Professional statisticians can provide advice. Otherwise the statistical conclusions for Figures 1 and especially Figures 5 and 6 cannot be trusted. Very puzzlingly, appropriate statistical tests seem to have them performed for Figures 7

and 8.

4. The experiments to evaluate the roles of mRNA localization in cell migration are important in terms of identifying its potential biological significance. The magnitudes of the effects seem rather disappointing (considerably less than two-fold). Perhaps more importantly, the so-called 3D migration experiments may not be measuring what the authors believe, i.e., "ability of cells to penetrate through 3D matrices." In addition, in Figure 8, the headings refer to "Invasion" through collagen and Matrigel, as do the Methods. In examining the invasion protocol modified and cited by the authors, the current study uses a substantially reduced concentration of Matrigel and an enormously reduced amount of collagen, making it seem quite unlikely that the cells are migrating through a 3D matrix. For example, unless there is a serious typographical error, the concentration of collagen is far below the amounts normally used for even migration on a tissue culture dish, and the Matrigel also seems to be quite dilute compared to the reference the authors cite. Can the authors demonstrate that these concentrations actually form gels permitting migration, as opposed to merely coating the pores in the membrane that can provide a matrix-coated tunnel through which cells migrate?

Also, do NIH 3T3 cells actually ever invade? For the latter point, it would be safest to refer to these experiments as another form of cell migration, not invasion.

5. Figure 7g is quite interesting in comparing the populations of RNAs found to be APC-dependent compared to those mislocalized by Pkp4 cUTR. Although the authors emphasize the high statistical significance of overlapping genes, a key fact seems to be that 66% of the RNAs do not coincide. This finding appears to imply that two-thirds of the populations are independently regulated. If this conundrum cannot be explained, the authors should at least discuss this issue in the Discussion.

6. The authors demonstrate quite clearly that the class of RNAs in protrusions entering 0.4 μm pores are quite different from those that enter 3 μm pores. The authors note that the tiny protrusions are "filopodia-like" but do they think there is some significance to the difference from the normal protrusions they study? An important aspect of the Abstract and paper is a distinction between high- and low-tractility protrusions, but the evidence for this distinction in terms of contractility in the different protrusion seems to be quite weak. There is currently increasing controversy about whether pMLC is a reliable measure of contractility, and making the conclusion that these protrusions are less contractile needs additional support. For example, if the authors wish to make this distinction, they need to show that inhibiting the tiny protrusions with blebbistatin does not have any effect. It might be safer for the authors to avoid trying to make this distinction between high- and low-tractility protrusions, unless they can provide convincing evidence. In fact, it could be argued that they may have identified interesting differences in RNA localization in filopodia versus lamellipodia. If so, one wonders whether their in situ hybridization analyses show differences in RNA localization in tiny filopodia compared to regular cell protrusions?

7. The reader is left in the dark about exactly what the RNA localization is doing biologically. Examining a previous paper indicates that knockdown of APC does not affect the formation of protrusions, though it is not clear here whether their other experimental manipulations can affect protrusions, e.g., numbers or activity. If the biological effect is instead a rather modest reduction in rate of migration, one wonders how that is occurring in terms of some mechanism? Although this paper would be considerably strengthened by an evaluation of whether the effect is on cell adhesions as might be inferred from the text, it might alternatively be important to examine for effects on migration by some method besides the clever but not conceptually airtight approach of competition used in this paper. Specifically, does knockdown of APC or some additional experimental manipulation that affects RNA localization, such as the authors' recently published change in mRNA localization by altering kinesin-1 or FUS aggregates, produce the same inhibition of cell migration?

8. Is it valid to conclude that the PDI measurements show that NIH 3T3 cells plated on a soft gel

show a different RNA distribution compared to cells on slightly stiffer or very stiff gels if the soft gel results in a cell that has very few total protrusions? A specific concern here is whether the PDI effects are solely due to matrix stiffness, or because they are actually affected by morphology of the cells, with few protrusions?

9. On page 12, the statement "We thus conclude that actomyosin contractility is required both for the formation of the detyrosinated-MT network as well as for APC-dependent RNA localization" is clearly correct for the microtubule network, but applying the word "required" for the RNA localization seems to be stretching the findings, which even though quite statistically significant, are changes that are unfortunately pretty small in terms of percentage changes of average PDI. Perhaps better-qualified caution in this conclusion could resolve this problem.

Additional, less-crucial points:

10. The authors should be commended for showing all their data points on multiple graphs, but the error bars indicating "95% confidence interval" seems quite puzzling, since the usual 95% confidence intervals for points with a normal distribution are much wider with much fewer outliers. Presumably the authors may be calculating this in some unusual manner, but it is rather distracting to see such narrow error bars, which may be merely something like standard error of the mean.

11. In the migration assays using transwell membranes, the authors believe they are using FBS as a "chemo-attractant" but the assay is conducted over a period of 48 hours. During this time, the FBS no doubt equilibrates with the upper chamber, and there is almost certainly no chemotactic gradient. The authors are instead probably studying chemokinesis, since a control using either 5% or 10% FBS in both the upper and lower chambers will likely give the same results in terms of numbers of cells crossing the membrane. Although this apparent problem negates the interpretation that this is chemotaxis, this reviewer feels that this fact is not a problem if the authors simply refer to this assay as a trans-filter migration assay.

12. It would be quite helpful if the authors would clarify in the Introduction or discussion whether the prior literature and this paper can allow conclusions about the role of local activation of the RhoA GTPase mentioned on lines 13-15 on page 4 involved in localizing actin and Arp2/3 is somehow related or unrelated to this paper's conclusion that regulates the distinct class of APC-dependent RNA localization mechanism.

13. Was there some reason for using the considerably lower number of samples (< half) for 3 conditions: Figure 3p right panel, 280 kPa; Figure 3d right panel, Y27632; and Figure 3i right panel: Rho Activator II?

14. Could the authors speculate on the significance of the increased perinuclear localization of polyA RNAs on stiffer substrates?

15. Can the authors provide any speculation concerning the finding that even though increased stiffness decreases PDI, but altering contractility does not, and yet activating Rho by its specific activator in Figure 3h appears to significantly reduce PDI?

16. The statement on page 19 "To our knowledge, this is the first report showing a requirement for a specific set of modified microtubules, namely detyrosinated microtubules, for transport of particular mRNAs" in light of their new paper found by this reviewer in PubMed reporting "...leading to a loss of detyrosinated glutamate (Glu)-microtubules (MTs; Glu-MTs) and an inability to support the localization of RNAs at protrusions."?

17. The statement on page 22 "Cell line has not been authenticated." Is rather disturbing. It is puzzling that the authors have not done so.

18. The Methods only list single siRNAs, so have the authors ruled out the possibility of off-target effects, since they are not using the standard 2 independent siRNAs or a cDNA rescue?

19. For some of the bar graphs, no statistical significance is evaluated. Doing so is not essential if there are obviously major differences, but the authors might consider doing so just for consistency.

20. In Figure 7A, what is the significance of the large dashed versus smaller dashed lines and solid lines?

21. Typos appear to be quite rare, but on page 22, line 12, "wiling" should probably read "wiping". There is an extra comma in line 14 of the Abstract. The authors should check that their use of the term "RNA-Seq" is consistent throughout the paper, e.g., compared to line 6 of the Abstract ("RNAseq"), etc. Page 8, line 1: "10 select" should read "10 selected"

21. In the Abstract, lines 13-15, the statement about "a competition-based approach to specifically and globally mislocalize APC-dependent RNAs..." seems quite confusing at first reading, since specifically and globally initially sounded mutually contradictory. Presumably, specifically refers to the use of a specific sequence and globally may refer to multiple RNAs. Perhaps removing "and globally" might help.

We would like to thank the reviewers for their thoughtful and very useful comments. We believe that addressing them has greatly improved the quality and clarity of the manuscript. Listed below are our point-by-point responses with references to text changes and additional data provided.

Reviewer #1, an expert in mRNA localization (Remarks to the Author):

Summary.

mRNAs are actively transported to many locations in diverse cell types and this process is evolutionarily conserved. Previous work from this group has demonstrated that a distinct set of mRNAs are localized to cell protrusions and that the localized RNAs are associated with the APC protein and Glu-microtubules. The current work expands on the previous studies to provide a genome-wide view of all the RNAs that require the APC protein for localization to protrusions. Additionally, the authors show that cell contractility influences the efficiency of mRNA localization through a pathway that regulates Glu-microtubule formation, although they are not able to establish a molecular link between these two pathways. Finally, the authors use an overexpression and competition experiment to show that mislocalization of a subset of APC-dependent RNAs results in defects in cell motility in 3D media. Ascribing a function for APC-dependent mRNA localization is a potentially very important addition to the literature. While the authors have provided a suggestion that mRNA localization is important for cell motility I believe that a few additional experiments are required to clearly demonstrate this important point. If the authors can address this issue and a one additional minor concern I would enthusiastically support publication and believe that this manuscript will be a valuable addition to the literature.

Major Points:

1. In Figure 8 the authors show that cells overexpressing Pkp4, Rab13, PkpA12, and PkpA12.1 cUTRs show reduced motility through 3D substrates. The magnitude of the motility effects (Figure 8) and effects on mRNA localization (Figure 7) are different for each of the constructs. The authors have not carefully characterized all aspects of each of these cUTR constructs. From the images provided in Figure 7 and Figure S7 it is clear that these constructs do not localize to protrusions with equal efficiency. I would like to see that authors quantitate localization efficiency (using PDI) for each of the cUTR constructs used in Figures 7 and 8.

We have provided PDI quantitations of all cUTR constructs (shown now in revised Fig. S7b). Indeed, the smaller deletion constructs, PkpA12 and PkpA12.1, localize less efficiently compared to the full-length UTR constructs. However, they do show significant localization compared to the negative control UTR constructs (HBB and PkpB). We think that, even though the localization of the smaller deletion constructs is not as efficient, they can still compete with the endogenous RNAs, because of their over-expression.

Regarding the magnitude of effects of the different cUTR constructs on RNA localization (Figure 7b) versus on motility (Figure 8), we don't think that a quantitative comparison is warranted. Firstly, we don't think that there is any basis to assume that these two cellular behaviors (RNA localization and motility) are linked by a simple linear relationship. Secondly, it is conceivable that different endogenous RNAs will be competed to different extents (based on their expression levels, affinities for factor(s) being titrated etc.). In the experiments presented in Figure 7b, we assess the effect on localization of two specific RNAs (Ddr2 and Kank2), but we do not know to what extent these, or other RNAs, which might be mislocalized to different degrees, are

responsible for the observed motility defects. Therefore, again, we think that a direct correlation should not necessarily be expected.

2. In Figure 7 the authors show that overexpression of Pkp4 cUTR results in the mislocalization of a substantial fraction of APC-dependent mRNAs. However, there are many mRNAs that show increased localization to protrusions and many APC-dependent mRNAs that are not mislocalized. In Figure 8 the authors show that overexpression of cUTR constructs results in motility defects, linking mRNA localization to cell motility, which is a very exciting finding. However, because the cUTR experiment clearly affects many mRNAs that are not regulated by APC it is important to have an orthogonal method to show that APC-dependent mRNA localization affects cell motility to confirm this finding. In Figure 1 the authors use APC siRNA to define APC-dependent mRNA localization. It would solidify the link between mRNA localization and cell motility presented in Figure 8 to look at cell motility in APC siRNA treated cells.

We have now provided cell motility data of cells treated with 2 different siRNAs against APC. Indeed, consistent with our model, we observe a reduction in migration velocity similar in magnitude to that observed in the case of cUTR-expressing cells (now shown in revised Fig. 8d, e). Migration defects upon APC loss have also previously been reported, however since loss of APC results in significant activation of Wnt-signaling, and of β -catenin transcriptional activity, it has been difficult to determine to what extent the role of APC in migration is mediated through transcriptional effects of activated β -catenin versus other APC functions that are independent of β -catenin activation. Importantly, we now show that while, as expected, siAPC-treated cells show a significant increase in baseline Wnt-signaling activation (i.e. in the absence of Wnt ligand), cUTR-expressing cells do not show any appreciable Wnt-signaling activation (shown in revised Fig. S9c). Therefore, cUTR expression affects the localization of APC-dependent RNAs without affecting other APC functions. These results, thus, support the model that at least some β -catenin-independent functions of APC in cell migration are mediated through its associated localized RNAs.

Minor points.

1. For each of the Figures showing PDI data the authors present a plot with points, mean and SE. In the Figure legends that authors state that each of the points is an individual cell and that the experiments have been repeated at least twice. I assume that that mean value is the mean of the pooled cells rather than the mean of the replicate experiments. I think that it would be more useful to present that mean value of the replicate experiments and the errors as the SD or SEM of the variation between biological replicates. The N of cells measured could then be stated in the legend. This presentation would give a more accurate depiction of the reproducibility of the experiment.

We have debated quite a bit about the way of presenting the data. We have settled on using the way presented in the figures because we think it is important to show the underlying variability of the cell population, without reducing it to a single average value (this point also seems to be supported by Reviewer #2; see comment 10 below).

Variability between replicates will affect in a similar way either the mean of the pooled cells (that we present) or the mean of the averages of each experiment (that the reviewer asks for). We thus don't think that any information is concealed by our way of presentation. To illustrate that, we include below, for all the graphs of Figure 3, a side-by-side comparison of our analysis versus the mean/SEM of the replicate experiments. (It would have indeed been possible that looking at the pooled cells could have skewed the results if a different number of cells was

included in the analysis from one replicate versus another. However, we take care to observe a similar number of cells in each replicate analyzed). Consequently, as seen below, the results are very similar regardless of the type of presentation. We therefore believe that our way of presentation is more transparent regarding the underlying distribution, without discounting any important information.

Reviewer #2, an expert in cell-matrix crosstalk and cell migration (Remarks to the Author):

This well-written manuscript provides evidence that the stiffness of extracellular matrix and cellular contractility can both control the localization of specific mRNAs to cellular protrusions, and that an mRNA subclass can be implicated in promoting cell migration using a novel cUTR competition approach that can mislocalize endogenous APC-dependent mRNAs. On the positive side, this manuscript provides new evidence linking extracellular matrix stiffness, cellular traction force, and actomyosin contractility to the specific localization of subclasses of mRNAs, as well as providing some insight into the biological significance of this phenomenon of localization to cellular protrusions; in terms of methods, the use of four independent replicates for RNA-seq is excellent, and the manipulations of Glu-microtubule levels are convincing. Apparent negative elements, however, include issues involving the interpretation of some of the localization data, uncertainty about the strength of some of the conclusions, and problems with statistical analyses. Many of the specific concerns may be possible to resolve by a reconsideration of some interpretations and rewording; a couple of the conclusions, however, may need revision. Nevertheless, this manuscript provides an intriguing advance in a research area that needs more of this type of exploration of the roles of local mRNA localization, its control, and its physiological importance. Consequently, even though a number of points are listed below, the intent is to strengthen this potentially valuable paper rather than to reject it.

1. There appears to be a basic conceptual problem in the interpretation of altered localization as quantified by PDI when polyA RNA PDI is used as the baseline for comparison, which occurs in Figures 3b and 3h. The interpretation becomes inconsistent with direct comparisons between Ddr2 RNA levels because there is sometimes an anomalous substantial decrease in polyA RNA PDI. Puzzlingly, such seemingly more-appropriate direct comparisons without this form of normalization are shown elsewhere, e.g., in Figure 3d and Figure 6b and 6d.

To be specific, Figure 3b would probably show no change in quantified Ddr2 RNA on substrates of different stiffness if these values were compared with each other. Instead, the authors perform comparisons with polyA RNA, which clearly moves centrally in cells. The biological or molecular significance of this latter movement is not clear, which should also be discussed. But the key point of this paper involves the localization of the APC-dependent subclass of RNAs, which clearly do not appear to change with stiffness. Strangely, the authors do perform this appropriate comparison in Figure 3d, where inhibition of contractility results in a significant alteration in the distribution of Ddr2 RNA. There appears to be a fundamental disconnect between sections b and d of this figure. Similarly puzzling is the failure to compare alterations in PDI consistently in Figures 3f versus 3h, where the effect of LPA may still be statistically significant, but perhaps not so clear with a RhoActivator II; these comparisons using more appropriate statistical analyses need to be performed [point #3], rather than trying to compare with polyA RNA, which is decreased by these Rho activators.

We appreciate the reviewer's observations. We had tried to address some of these in a brief note mentioned in the legend of Figure 3. We have now transferred this to the main text and have clarified considerations regarding PDI analysis in the Methods section.

Our analysis compares the distribution of a specific RNA to a control RNA in order to assess preferential localization to the periphery (e.g. between Ddr2 and polyA RNA). We are also trying

to see whether different treatments affect the peripheral distribution of an RNA. What the reviewer points out as our inconsistency in performing the right comparisons stems from two issues, which in some cases limit our ability to compare distributions across different conditions. As detailed below these mostly apply when cells show significantly different spreading areas. We don't think that these considerations limit or question the conclusions reached in the manuscript.

Firstly, (as was stated previously in the legend of Figure 3), prior to PDI calculation, we subtract the signal within the nuclear area because: a) we are interested in assessing distributions in the cytoplasm and b) we know that our oligodT probes, which are used to visualize polyA RNA, show non-specific staining of DNA in the nucleus (based on control experiments using RNase or DNase-treated cells). This nuclear subtraction introduces a limitation in comparing PDI values between cells with very different spreading areas. That is because in cells with smaller spreading area (such as for those on soft substrates; spreading areas are shown in Figure S5a) a relatively larger proportion of the cellular area is covered by the nucleus. Subtracting that signal leads to an increase in PDI values (PDI=1 if the signal is completely diffuse throughout the cell; when omitting the central nuclear part, a completely diffuse cytoplasmic signal shows a PDI>1). While this does not affect the comparison of cytoplasmic distribution of polyA RNA to another RNA within the same cell, it nevertheless prevents a direct comparison to cells with larger spreading areas, where the nuclear signal covers a smaller proportion of the overall cellular area (compare nuclear outlines to cell outlines in Figure 3b). In these more spread cells the PDI value of a diffuse cytoplasmic signal would be closer to 1 after subtraction of the nuclear signal. For this reason, the PDI values on substrates of different stiffness (Figure 3b) cannot be directly compared and the similarity, for example, of the absolute Ddr2 PDI values on soft and stiff substrates cannot be taken to reflect an equal peripheral distribution.

Secondly, as the reviewer points out, another difference between less or more spread cells is the overall distribution of polyA RNA, which appears more central as the spreading area increases. However, the more prominent perinuclear accumulation of polyA RNA is not a result of the RNA moving centrally, but rather of polyA RNA reflecting the overall cytoplasmic cell volume. As depicted in the schematic below, which shows side views of different cells, cells with small spreading area are overall rounder and have a relatively uniform distribution of the cytoplasmic volume across the cell body. We image cells by taking serial z-slices throughout the cell volume and projecting those onto a 2-dimensional image. Since polyA RNA is relatively diffuse in the cytoplasm, projected images of rounder cells show similar polyA intensity throughout the cytoplasm. By contrast, when cells have a larger spreading area, protrusive regions are much thinner with most of the cytoplasm being present around the nucleus (see schematic below). Therefore, when z-stacks of these cells are projected on a 2D image, polyA signal appears more intense around the nucleus reflecting the overall higher amount of cytoplasm.

Because of these two factors we do not think it is valid to compare directly PDI values between cells with very different spreading areas. When conditions have a significant effect on cellular

spreading area (such as in Figure 3b and 3h; corresponding changes in cell area are shown in Figure S5a and S5e) we do not compare PDIs between different conditions, but rather the PDIs of an RNA with its corresponding polyA control to show whether an RNA shows a different distribution than the overall polyA RNA population. We compare PDIs across different conditions (e.g. Figures 3d, 3f, 6b, 6d) only when treatments don't affect significantly the cellular area, which can also be seen by the lack of change in the corresponding polyA RNA PDI values.

2. A related concern is that the wrong comparisons in Figure 5 and Figure 6 were performed in terms of statistical analyses, even though this reviewer thinks that the results might potentially help to support the authors' interpretations. Specifically, in Figure 5b, the correct comparison is performed, yet the same comparison is missing from Figure 5c, in which the comparison is suddenly with polyA RNA, which is an inconsistent analysis compared with the preceding figure. In Figure 6b, the most important analysis needs to be between the si-Control values and the si-TTL values, where it appears that there will be the hoped-for statistically significant rescue of the +blebbistatin condition by si-TTL, at least at a statistically significant level. Similarly, the rescue by mDia in the presence of blebbistatin is not compared.

These were inadvertent errors and have been corrected.

3. An important problem occurs in multiple important figures with respect to the choice of the method of statistical analysis (which is likely to result in apparent levels of statistical significance that are actually not correct) by using Student's t-test for experiments in which there are multiple variables. One cannot simply choose a pair of variables to compare without applying an appropriate statistical correction to the t-test, such as the Bonferroni correction. The alternative is to perform ANOVA with a statistically appropriate post-test. Professional statisticians can provide advice. Otherwise the statistical conclusions for Figures 1 and especially Figures 5 and 6 cannot be trusted. Very puzzlingly, appropriate statistical tests seem to have them performed for Figures 7 and 8.

Sorry for the oversight. P-values are now presented based on ANOVA analysis with Bonferroni's multiple comparisons test for Fig 1e, Fig 5, Fig 6, Fig 7b. Statistics have also been included in Fig S2 and the new Supplementary Fig S3a. There is no change in the statistical differences or any of the conclusions previously described.

4. The experiments to evaluate the roles of mRNA localization in cell migration are important in terms of identifying its potential biological significance. The magnitudes of the effects seem rather disappointing (considerably less than two-fold). Perhaps more importantly, the so-called 3D migration experiments may not be measuring what the authors believe, i.e., "ability of cells to penetrate through 3D matrices." In addition, in Figure 8, the headings refer to "Invasion" through collagen and Matrigel, as do the Methods. In examining the invasion protocol modified and cited by the authors, the current study uses a substantially reduced concentration of Matrigel and an enormously reduced amount of collagen, making it seem quite unlikely that the cells are migrating through a 3D matrix. For example, unless there is a serious typographical error, the concentration of collagen is far below the amounts normally used for even migration on a tissue culture dish, and the Matrigel also seems to be quite dilute compared to the reference the authors cite. Can the authors demonstrate that these concentrations actually form gels permitting migration, as opposed to merely coating the pores in the

membrane that can provide a matrix-coated tunnel through which cells migrate? Also, do NIH 3T3 cells actually ever invade? For the latter point, it would be safest to refer to these experiments as another form of cell migration, not invasion.

Indeed, NIH/3T3 cells do not move as efficiently as cancer cells through 3D matrices. That is why we had to use 3D gels of lower concentrations of ECM than usual. (There was, however, also a typo error in the collagen concentration. It should be 300 μ g/ml or 0.3mg/ml instead of the stated 300ng/ml). This is still lower than the concentration used in other reports. However, we had used reflection microscopy to verify that, even at that lower concentration, collagen forms 3-dimensional gels. This is shown in the images attached below, which compare a usual collagen gel (at 3 mg/ml) with the lower concentration (0.3 mg/ml) used in our experiments. We observe that, at 3 mg/ml, collagen forms apparently denser gels, but it can still form 3D gels at the lower concentration we use. These gels can support embedded cells (the brighter foci in the 0.3 mg/ml images are embedded cells. This is for illustration purposes; in the actual experiments the cells are placed on top of the gels). Matrigel does not produce a clear reflection pattern, so we couldn't use similar experiments to test its actual appearance, but we do observe that it forms a gel of detectable thickness. Therefore, we believe that in our assays cells are moving through a 3-dimensional matrix.

Nevertheless, we agree with the reviewer that NIH 3T3 cells likely don't invade, in the sense that they probably don't remodel or digest the ECM, but rather move through it. We now explain this more explicitly in the text and have removed any reference to 'invasion' and now refer to it more consistently as 'movement (or migration) through a 3-dimensional matrix'.

5. Figure 7g is quite interesting in comparing the populations of RNAs found to be APC-dependent compared to those mislocalized by Pkp4 cUTR. Although the authors emphasize the high statistical significance of overlapping genes, a key fact seems to be that 66% of the RNAs do not coincide. This finding appears to imply that two-thirds of the

populations are independently regulated. If this conundrum cannot be explained, the authors should at least discuss this issue in the Discussion.

We don't expect that our approach will result each time in an unambiguous identification of all protrusion-enriched RNAs. Variabilities could be introduced by the quality of the fractionation, the quality and biases introduced by the RNA-seq procedure and the specific cutoffs used. To minimize any biases, the samples of a given experiment are fractionated and sequenced in parallel. This allows us to identify protrusion-enriched RNAs, and associated effects on them, with high confidence within an experiment (i.e. for the sample sets presented in Figure 1 and Figure 7). These sources of variability, however, can have a larger influence when comparing datasets from samples acquired and sequenced separately. To illustrate this point, for example, using the same cutoffs, 914 RNAs were identified as protrusion-enriched in the control cells of Figure 1, whereas 586 were identified in the control cells of Figure 7. We, therefore, think it is not surprising that we see a partial overlap when comparing the APC-dependent RNAs to those mislocalized by the Pkp4 cUTR. We cannot exclude that there are some independently-regulated RNAs, but we had not emphasized this point since we cannot distinguish those from ones resulting from experimental variability. We now make a brief mention of that in the text.

6. The authors demonstrate quite clearly that the class of RNAs in protrusions entering 0.4 μm pores are quite different from those that enter 3 μm pores. The authors note that the tiny protrusions are “filopodia-like” but do they think there is some significance to the difference from the normal protrusions they study? An important aspect of the Abstract and paper is a distinction between high- and low-tractility protrusions, but the evidence for this distinction in terms of contractility in the different protrusion seems to be quite weak. There is currently increasing controversy about whether pMLC is a reliable measure of contractility, and making the conclusion that these protrusions are less contractile needs additional support. For example, if the authors wish to make this distinction, they need to show that inhibiting the tiny protrusions with blebbistatin does not have any effect. It might be safer for the authors to avoid trying to make this distinction between high-and low-tractility protrusions, unless they can provide convincing evidence. In fact, it could be argued that they may have identified interesting differences in RNA localization in filopodia versus lamellipodia. If so, one wonders whether their in situ hybridization analyses show differences in RNA localization in tiny filopodia compared to regular cell protrusions?

Our intent was to show that different sets of RNAs become enriched in different types of protrusions (i.e. the small and large protrusions presented in Figure 2). The presence of actin bundles in the large protrusions and the reduced levels of both total myosin as well as pMLC levels in the small protrusions indicated to us that these types of protrusions might differ in their degree of contractility. Based on this, we then directly tested the role of contractility and the underlying mechanism in APC-dependent RNA localization (Figures 3-6). We agree that we cannot draw a definitive conclusion about the contractility status of the small protrusions. This was stated more forcefully than intended in the abstract, because of trying to limit the word length. We have now reworded the text to avoid any misconceptions.

We have tried quite a bit to determine what the small protrusions might correspond to in a migrating cell on a 2D surface, but have been unable to reach any definite conclusions.

7. The reader is left in the dark about exactly what the RNA localization is doing biologically. Examining a previous paper indicates that knockdown of APC does not affect the formation of protrusions, though it is not clear here whether their other

experimental manipulations can affect protrusions, e.g., numbers or activity. If the biological effect is instead a rather modest reduction in rate of migration, one wonders how that is occurring in terms of some mechanism? Although this paper would be considerably strengthened by an evaluation of whether the effect is on cell adhesions as might be inferred from the text, it might alternatively be important to examine for effects on migration by some method besides the clever but not conceptually airtight approach of competition used in this paper. Specifically, does knockdown of APC or some additional experimental manipulation that affects RNA localization, such as the authors' recently published change in mRNA localization by altering kinesin-1 or FUS aggregates, produce the same inhibition of cell migration?

In response also to point #2 of reviewer #1, we have now provided cell motility data of cells treated with 2 different siRNAs against APC. Indeed, consistent with our model, we observe a reduction in migration velocity similar in magnitude to that observed in the case of cUTR-expressing cells (now shown in revised Fig. 8d, e). Migration defects upon APC loss have also previously been reported, however since loss of APC results in significant activation of Wnt-signaling, and of β -catenin transcriptional activity, it has been difficult to determine to what extent the role of APC in migration is mediated through transcriptional effects of activated β -catenin versus other APC functions that are independent of β -catenin activation. Importantly, we now show that while, as expected, siAPC-treated cells show a significant increase in baseline Wnt-signaling (i.e. even in the absence of Wnt ligand), cUTR-expressing cells do not show any appreciable Wnt-signaling activation (shown in revised Fig. S9c). Therefore, cUTR expression can affect the localization of APC-dependent RNAs without affecting other APC functions. These results, thus, support the model that at least some β -catenin-independent functions of APC in cell migration are mediated through its associated localized RNAs.

We believe that RNA localization is part of a mechanism used to modulate the efficiency of cell migration and not an integral component of the cell's ability to move. Hence, the magnitude of the observed effects is small relative to treatments that would be expected to affect fundamental aspects of cell migration. We are, of course, following up on the exact cellular features affected by localized RNAs, but an in depth understanding of the mechanisms will require first the identification of individual RNAs affecting specific aspects of cell migration, and we think this is beyond the scope of the current manuscript.

8. Is it valid to conclude that the PDI measurements show that NIH 3T3 cells plated on a soft gel show a different RNA distribution compared to cells on slightly stiffer or very stiff gels if the soft gel results in a cell that has very few total protrusions? A specific concern here is whether the PDI effects are solely due to matrix stiffness, or because they are actually affected by morphology of the cells, with few protrusions?

We understand this concern, and as stated in the response to point #1, this partly accounts for the limitation in directly comparing PDIs across different substrates. However, we think that our experiments, looking at cells on substrates of the same stiffness but treated with different inhibitors or siRNAs, suggest that the morphology of the cells does not account for the changes in PDI values. Specifically, for example, cells on 5 kPa substrates (Figure 3d) treated with blebbistatin or Y27632 show similar morphologies (if anything, blebbistatin-treated cells often have even longer protrusions than control cells), but show reduced PDI values. Also, cells on 1kPa substrates (Figure 1h) treated with LPA or Rho Activator do not exhibit protrusions similar to the ones seen on the stiffer substrates, yet they show increased PDI values.

9. On page 12, the statement “We thus conclude that actomyosin contractility is required both for the formation of the deetyrosinated-MT network as well as for APC-dependent RNA localization” is clearly correct for the microtubule network, but applying the word “required” for the RNA localization seems to be stretching the findings, which even though quite statistically significant, are changes that are unfortunately pretty small in terms of percentage changes of average PDI. Perhaps better-qualified caution in this conclusion could resolve this problem.

We have toned down the sentence.

Additional, less-crucial points:

10. The authors should be commended for showing all their data points on multiple graphs, but the error bars indicating “95% confidence interval” seems quite puzzling, since the usual 95% confidence intervals for points with a normal distribution are much wider with much fewer outliers. Presumably the authors may be calculating this in some unusual manner, but it is rather distracting to see such narrow error bars, which may be merely something like standard error of the mean.

95% confidence intervals are included in graphs generated from the raw values in GraphPad Prism software. The intervals are likely narrow due to the large n numbers.

11. In the migration assays using transwell membranes, the authors believe they are using FBS as a “chemo-attractant” but the assay is conducted over a period of 48 hours. During this time, the FBS no doubt equilibrates with the upper chamber, and there is almost certainly no chemotactic gradient. The authors are instead probably studying chemokinesis, since a control using either 5% or 10% FBS in both the upper and lower chambers will likely give the same results in terms of numbers of cells crossing the membrane. Although this apparent problem negates the interpretation that this is chemotaxis, this reviewer feels that this fact is not a problem if the authors simply refer to this assay as a trans-filter migration assay.

That’s a valid point. We now state that ‘FBS was added in the lower chamber to induce cell motility’ and have removed references to chemotactic migration.

12. It would be quite helpful if the authors would clarify in the Introduction or discussion whether the prior literature and this paper can allow conclusions about the role of local activation of the RhoA GTPase mentioned on lines 13-15 on page 4 involved in localizing actin and Arp2/3 is somehow related or unrelated to this paper’s conclusion that regulates the distinct class of APC-dependent RNA localization mechanism.

We had not expanded on that topic because not much is known about how RhoA affects localization of actin or the Arp2/3 RNAs. We don’t observe localization of these RNAs in our system and we don’t think we can draw any conclusions about potential connections, or not, between our observations and these prior reports.

13. Was there some reason for using the considerably lower number of samples (< half) for 3 conditions: Figure 3p right panel, 280 kPa; Figure 3d right panel, Y27632; and Figure 3i right panel: Rho Activator II?

For the 280 kPa gels, because of the high stiffness, it is quite difficult to image cells that can be reliably analyzed. In the other cases, there was no particular reason. Based on the extent and variance of the observed differences, the sample sizes were sufficient for drawing conclusions. Given that the traction force microscopy experiments are quite laborious both in their execution and analysis, we didn't opt to further increase the sample size.

14. Could the authors speculate on the significance of the increased perinuclear localization of polyA RNAs on stiffer substrates?

The more prominent perinuclear accumulation of polyA RNA is not a result of the RNA moving centrally, but rather of polyA RNA reflecting the overall cytoplasmic cell volume. Please see response to point #1 for a more detailed explanation.

15. Can the authors provide any speculation concerning the finding that even though increased stiffness decreases PDI, but altering contractility does not, and yet activating Rho by its specific activator in Figure 3h appears to significantly reduce PDI?

We are assuming that the reviewer refers to the PDI values of the polyA RNA. If that is the case, then these differences are due to the PDI of the polyA RNA being affected by changes in the cell surface area (again, please see response to point #1 for a more detailed explanation). Briefly, polyA RNA PDI is reduced (figures 3b and 3h) when there are significant changes in cell spreading area (shown in Fig S5). Treatments that don't significantly alter the cell area do not affect poly RNA PDI. As stated in the response to point #1, this is the reason we don't compare PDIs across conditions when cell surface areas are also affected.

16. The statement on page 19 "To our knowledge, this is the first report showing a requirement for a specific set of modified microtubules, namely detyrosinated microtubules, for transport of particular mRNAs" in light of their new paper found by this reviewer in PubMed reporting "...leading to a loss of detyrosinated glutamate (Glu)-microtubules (MTs; Glu-MTs) and an inability to support the localization of RNAs at protrusions."?

This is because the two manuscripts were being submitted in parallel. We have modified the sentence to point out that APC-dependent RNAs are the first group of RNAs shown to require modified microtubules, and have referenced our recent publication supporting the same conclusion.

17. The statement on page 22 "Cell line has not been authenticated." Is rather disturbing. It is puzzling that the authors have not done so.

The statement was quite explicit for the purposes of full disclosure. We have used mouse cells in our experiments and, to our knowledge, at least up until a few months ago, there was no standardized, commercially available method for mouse cell line authentication. NIST and ATCC apparently have partnered to establish a Mouse Cell Line Authentication Consortium to test the effectiveness of mouse STR markers, create a public STR profile database for mouse cell lines and publish consensus standards for mouse cell line authentication (Almeida JL et al (2016) PLOS Biology) (<https://www.nist.gov/programs-projects/cell-line-identification-and-authentication>). As far as we know, this work is still ongoing and there are no commercial kits available or standard methods in place. We thus could not authenticate the cell line used.

Nevertheless, we don't believe this can impact in any way on the results reported since we are describing basic cellular mechanisms. Furthermore, we would like to point out that because of the multiple RNA-Seq analyses performed on these cells we know that they are mouse cells and have verified the expression status of all relevant factors.

18. The Methods only list single siRNAs, so have the authors ruled out the possibility of off-target effects, since they are not using the standard 2 independent siRNAs or a cDNA rescue?

We have included experiments testing the effect of APC knockdown using a different siRNA. Due to the cost, we have tested the effect on RNA localization, not through RNA-Seq, but through nanoString analysis detecting several APC-dependent and APC-independent RNAs. Consistent with our prior conclusions, we find that APC KD reduces the enrichment at protrusions of RNAs that we had previously identified as APC-dependent, but not of RNAs that we had identified as APC-independent (revised Fig S3a,b). The two different siRNAs against APC also have identical effects on cell migration (revised Fig 8d, e).

We have also tested additional siRNAs against TTL and myosin heavy chain (Myh9) (revised Fig S6a and d). Again, the results are similar to those previously described, indicating that indeed the observed effects result from loss of expression of the targeted genes.

19. For some of the bar graphs, no statistical significance is evaluated. Doing so is not essential if there are obviously major differences, but the authors might consider doing so just for consistency.

Statistical tests have been included.

20. In Figure 7A, what is the significance of the large dashed versus smaller dashed lines and solid lines?

The different line types are meant to indicate UTRs of different sequences. This has been clarified in the figure legend.

21. Typos appear to be quite rare, but on page 22, line 12, "wiling" should probably read "wiping"

There is an extra comma in line 14 of the Abstract.

The authors should check that their use of the term "RNA-Seq" is consistent throughout the paper, e.g., compared to line 6 of the Abstract ("RNAseq"), etc.

Page 8, line 1: "10 select" should read "10 selected"

Thanks, these have been corrected.

21. In the Abstract, lines 13-15, the statement about "a competition-based approach to specifically and globally mislocalize APC-dependent RNAs..." seems quite confusing at first reading, since specifically and globally initially sounded mutually contradictory. Presumably, specifically refers to the use of a specific sequence and globally may refer to multiple RNAs. Perhaps removing "and globally" might help.

Sentence has been edited.

Reviewers' Comments:

Reviewer #1:

Remarks to the Author:

protein for localization to different types of cellular protrusions. They also provide evidence that there is a link between cellular contractility and the extent of mRNA localization. Finally, the authors now use two different complementary approaches to show that failure to localize some APC-dependent mRNAs to cellular protrusions results in defects in cellular migration through 3D substrates. In this revision the authors have addressed all the points that I raised in my previous review and added experiments that I believe significantly strengthen the results of the paper. I now believe that this manuscript provides important new insight into the process of mRNA localization and is a valuable addition to the literature.

Reviewer #2:

Remarks to the Author:

This resubmitted manuscript has resolved, within reason, all of the concerns raised in the original review.

There is one minor typographical error: on page 18, lines 19-20, the word "to" is missing from the comment "it has been difficult determine to what extent"

Acceptance for publication is now recommended.

We are delighted that the reviewers think the current version of the manuscript is suitable for publication and we appreciate their very useful contributions.
We have corrected the typo error mentioned.

REVIEWERS' COMMENTS:

Reviewer #1 (Remarks to the Author):

protein for localization to different types of cellular protrusions. They also provide evidence that there is a link between cellular contractility and the extent of mRNA localization. Finally, the authors now use two different complementary approaches to show that failure to localize some APC-dependent mRNAs to cellular protrusions results in defects in cellular migration through 3D substrates. In this revision the authors have addressed all the points that I raised in my previous review and added experiments that I believe significantly strengthen the results of the paper. I now believe that this manuscript provides important new insight into the process of mRNA localization and is a valuable addition to the literature.

Reviewer #2 (Remarks to the Author):

This resubmitted manuscript has resolved, within reason, all of the concerns raised in the original review.

There is one minor typographical error: on page 18, lines 19-20, the word "to" is missing from the comment "it has been difficult determine to what extent"

Acceptance for publication is now recommended.